# 🧑‍🎓 AcadReason: Exploring the Limits of Reasoning Models with Academic Research Problems

**Xin Gui**[1,2,3*]   **King Zhu**[2,3*]   **JinCheng Ren**[3,4*]   **Qianben Chen**[3]   **Zekun Moore Wang**[5]
**Yizhi Li**[6]   **Xinpeng Liu**[10]   **Ren Wenli**[3]   **Linyu Miao**[3]   **Tianrui Qin**[3]
**Ziqi Shu**[3]   **He Zhu**[3]   **Dingfeng Shi**[3]   **Jiaheng Liu**[7]   **Yuchen Eleanor Jiang**[3]
**Minghao Liu**[8†]   **Ge Zhang**[2,9†]   **Wangchunshu Zhou**[3†]

[1]BUPT   [2]MAP   [3]OPPO   [4]Hohai University   [5]Kuaishou   [6]The University of Manchester
[7]Nanjing University   [8]2077AI   [9]ByteDance   [10]Peking University
 wcszhou@outlook.com, gezhang@umich.edu, dreamforever.liu@gmail.com

## Abstract

In recent years, the research focus of large language models (LLMs) and agents has shifted increasingly from demonstrating novel capabilities to complex reasoning and tackling challenging tasks. However, existing evaluations focus mainly on math/code contests or general tasks, while existing multi-domain academic benchmarks lack sufficient reasoning depth, leaving the field without a rigorous benchmark for high-level reasoning. To fill this gap, we introduce the AcadReason benchmark, designed to evaluate the ability of LLMs and agents to acquire and reason over academic knowledge. It consists of 50 expert-annotated academic problems across five high-reasoning domains, including computer science, economics, law, mathematics, and philosophy. All questions are sourced from top-tier publications in recent years and undergo rigorous annotation and quality control to ensure they are both challenging and answerable. We conduct systematic evaluations over 10 mainstream LLMs and agents. The results show that most LLMs scored below 20 points, with even the cutting-edge GPT-5 achieving only 16 points. While agents achieved higher scores, none exceeded 40 points. This demonstrates the current capability gap between LLMs and agents in super-intelligent academic research tasks and highlights the challenges of AcadReason. The code and data for the AcadReason benchmark are available at https://github.com/OPPO-PersonalAI/Acadreason-benchmark.

## 1 Introduction

The ability to reason effectively is a cornerstone of advanced artificial intelligence, enabling systems to tackle complex problems across diverse domains. Recent advancements in large language models (LLMs), exemplified by models such as OpenAI's o3 (OpenAI, 2024b), have demonstrated significant strides in reasoning capabilities. These models leverage techniques like inference-time scaling and learning-to-reason, showcasing robust performance across reasoning tasks (Ke et al., 2025).

However, as reasoning LLMs continue to evolve, limitations in existing reasoning benchmarks—such as MMLU-Pro (Wang et al., 2024), GPQA (Rein et al., 2023) and SuperGPQA (Team et al., 2025)—have become apparent. These benchmarks, designed for simpler tasks like arithmetic, algebra, grade-school knowledge, or commonsense reasoning, are becoming outdated and saturated, failing to capture the trends of advanced reasoning.

For example, GAIA (Mialon et al., 2024) assesses LLMs' general agentic abilities through real-world questions, while PaperBench (Starace et al., 2025) challenges LLMs to replicate 20 ICML machine

---

* Equal contribution.
† Corresponding Author.

learning papers, testing their abilities in coding, debugging, paper comprehension, and scientific reasoning. A more detailed comparison to other benchmarks can be found in E.4.

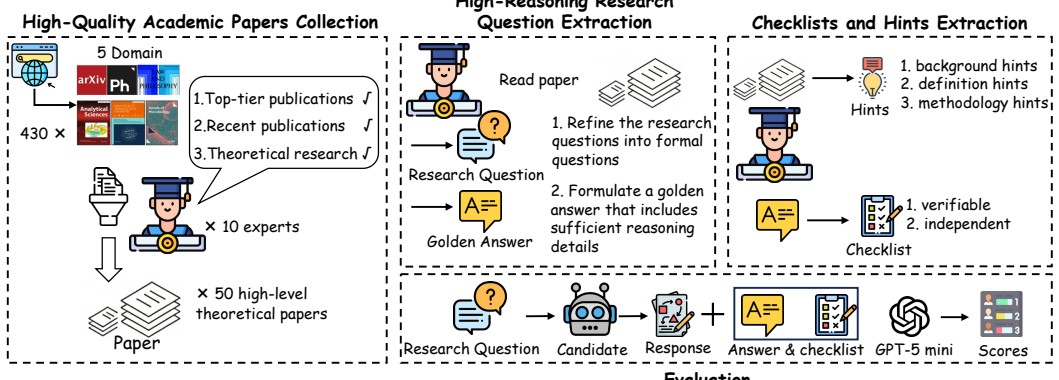

Figure 1: Overview of the ACADREASON benchmark construction and evaluation pipeline. It consists of three stages: **(1) High-Quality Academic Papers Collection** – experts filter 430 papers across 5 domains into 50 top-tier theoretical works; **(2) High-Reasoning Research Question Extraction** – research questions are refined into formal queries with golden answers containing sufficient reasoning; **(3) Checklists and Hints Extraction** – background, definition, and methodology hints are provided together with verifiable, independent checklists. For evaluation, candidate responses are compared against golden answers and checklists, and GPT-5 mini assigns final scores.

Despite these advancements, these benchmarks often lack either domain breadth - failing to comprehensively cover fields like science and humanities - or depth of difficulty - missing the professional rigor, timeliness, and complexity required for cutting-edge reasoning tasks. To address these shortcomings, we propose ACADREASON, a benchmark designed evaluate the academic-level reasoning abilities of LLM and agent.

As shown in Figure 1, our approach involves extracting knowledge and synthesizing high-quality reasoning data from diverse, authoritative, and timely academic literature spanning domains such as philosophy, statistics, mathematics, economics, computer science, among others. Specifically, based on publication date and top-tier journal status, we select 430 papers from leading journals. From each paper, we extract only one research question, with the corresponding golden answer designed to cover the full scope of the paper's contributions, thereby making each task demanding in workload and reasoning depth. Building on the extracted information, we further develop a scoring checklist and hints, providing more detailed evaluation rules and experiments. Ultimately, we compile a total of 50 high-quality research question, forming the ACADREASON benchmark. For evaluation, we adopt LLM-as-Judge as our evaluation method and utilize GPT-5-mini as the judge model, which conducts assessments based on detailed scoring criteria and a checklist.

Our experimental results demonstrate that ACADREASON provides challenging tasks for LLMs and agents. Even the latest and most powerful model, GPT-5, achieves only 16 points in pass rate and 40.6 points in checklist score. Furthermore, we find that reasoning models outperformed general models, with DeepSeek-R1 attaining a checklist score of 23.8, higher than DeepSeek-V3's 15.9 points. We also test cutting-edge agents, OAgents achieve the highest score of 34 points among all models and frameworks, demonstrating agents' strong capability in solving research problems, though significant room for improvement remains. Additionally, we introduce hints to investigate the impact of different types of knowledge during problem-solving. The experimental results indicate that the incorporation of hints, as supplementary information, positively contributes to model performance, with methodology hints yielding the most significant gains. This suggests that, compared to simple and easily accessible background information, the ACADREASON benchmark places greater emphasis on evaluating LLMs' mastery of methods.

Our contributions are as follows:

- We introduce ACADREASON benchmark, which provides multi-domain evaluation of LLMs' high-level reasoning abilities, and introduces challenges to existing models in terms of both knowledge and reasoning capability.

- We evaluate SOTA LLMs and Agents, Our testing experiments demonstrate that general models underperform on ACADREASON while reasoning models and agents exhibit stronger but still improvable performance, validating the dataset's challenge level and reasoning-centric design.

- We provide comprehensive and detailed evaluation metrics, along with different types of knowledge hints. This offers insights for uncovering the potential of LLMs and Agents, as well as guiding future improvement directions.

## 2 RELATED WORK

**Large Reasoning Models and Agent** With the release of Deepseek-R1 (Guo et al., 2025b) and OpenAI's o1 (OpenAI et al., 2024) model, LRMs(Large Reasoning Models) have demonstrated remarkable performance in areas such as inference and academic competitions. Deepseek-R1 (Guo et al., 2025b) extends the model's reasoning chain through reinforcement learning approach, achieving impressive results. Qwen3 (Yang et al., 2025) offers a hybrid reasoning mode alongside a default mode, providing more flexible thinking strategies. Although LRMs possess exceptional capabilities in reasoning, they are constrained by their limited internal knowledge. The Agent Framework (Zhu et al., 2025a;b; Team, 2025a; Fang et al., 2025; Qin et al., 2025) builds upon the foundational abilities of LRMs and extends them with corresponding tools, enhancing the model's capacity to acquire external knowledge. OAgents (Zhu et al., 2025a) conduct a systematic empirical study on the GAIA benchmark and BrowseComp, achieving outstanding performance. MiroFlow (Team, 2025a) constructs its agent framework based on MCP and has achieved state-of-the-art results on multiple leaderboards.

**Reasoning Benchmark.** Evaluating advanced reasoning capabilities remains a central challenge in the development of language models. Benchmarks such as arXivBench (Li et al., 2025a) and Paper-Bench (Starace et al., 2025) have been designed to assess the research-related reasoning abilities of LLMs. arXivBench requires LLMs to generate accurate paper names and corresponding links, while PaperBench evaluates models' ability to reproduce ICML papers. DeepResearch Bench (Du et al., 2025) assembles multi-domain tasks to evaluate LLMs' research-oriented reasoning. GAIA(Mialon et al., 2024) presents real-world challenges that require models to demonstrate proficient tool usage, web search capabilities, and reasoning. BrowseComp (Wei et al., 2025) places greater emphasis on web search and the ability to synthesize information from multiple web pages. However, existing benchmarks are limited in two key aspects: some lack breadth of coverage, being overly focused on math and coding at the expense of fields like science and humanities, while others lack depth of reasoning, testing only superficial information integration rather than advanced, professional knowledge. In contrast, our work bridges this gap by integrating both dimensions, presenting a novel and comprehensive challenge to evaluate the ability of LLMs and Agents to tackle cutting-edge academic research questions.

## 3 ACADREASON BENCHMARK

In this section, we introduce the ACADREASON benchmark, which focuses on measuring the cutting-edge reasoning capabilities of LLMs. Our human annotation process encompasses multiple stages, including data collection, question extraction, and quality assurance, to ensure the quality and challenge level of the questions. To establish a comprehensive evaluation framework, we incorporate hints and checklists based on questions and answers, thereby constructing a robust evaluation methodology with corresponding metrics (specific data can be found in Appendix G ).

### 3.1 TASK SPECIFICATION

In ACADREASON, LLMs and agents serve as candidates and are tasked in the role of a researcher. They are required to solve complex research questions extracted from high-level theoretical articles without access to the original text, relying either on internal knowledge or utilizing search tools to obtain additional information. Unlike simple information retrieval and integration, ACADREASON simulates real-world research scenarios, demanding that the models not only possess cutting-edge academic knowledge but also demonstrate deep reasoning capabilities.

Formally, each task in ACADREASON benchmark contains such atomic fields:

- **Question**: Each question is a research question constructed from the selected paper, which is self-contained, comprising (a) a specific problem from the paper and (b) the minimal background necessary for comprehension.
- **Hints**: Supporting information provided to the candidate model. To analyze the impact of different information types, hints are divided into three categories:
    - **Background Hints**: background knowledge and related work.
    - **Definition Hints**: key concepts and terminology introduced in the paper.
    - **Methodology Hints**: theoretical tools required for reasoning and proof.
- **Checklist**: Expert-designed checkpoints that capture key milestones in the reasoning process (e.g., logical steps or critical facts). Unlike static checklists in prior work, ours are dynamic, tailored to each question, and adapt in length to problem complexity.
- **Golden Answer**: A complete solution trajectory that fully satisfies all checklist requirements, covering background, definitions, derivations, and conclusions.

## 3.2 DATA ANNOTATION

Task construction in the ACADREASON benchmark follows strict principles to ensure quality, clarity, and alignment with high-information, high-reasoning challenges. Our data annotation pipeline consists of three components: 1. Collection of high-quality academic papers as raw material. 2. Extraction of high-reasoning question-answer pairs. 3. Development of checklists and hints based on golden answers. The annotation guideline can be found in Appendix F

**High-Quality Academic Papers Collection**   To ensure the challenging nature of the questions in ACADREASON, we design a meticulous data selection protocol. First, based on criteria including publication date and top-tier journal status, we collect 430 eligible papers from various leading journal websites. These papers cover a wide range of domains and exhibit diverse domain-specific logics, though not all are necessarily suitable for conversion into question-answer format. Annotation experts are instructed to carefully review and filter these articles according to the following principles: 1. whether they contain challenging reasoning questions, 2. whether they consist of purely theoretical content.

**High-Reasoning Research Questions Extraction**   Based on the collected high-quality papers, annotators are required to extract high-reasoning questions and golden answers from them. First, annotators read the entire paper and identify its main contributions and core research questions. Then, they refine the research questions into formal questions that must meet the requirements of being Comprehensive and Challenging. Finally, based on the question and the full content of the paper, the annotators formulate a golden answer that includes sufficient reasoning details—such as definitions, formulas, key concepts, and derivations—while also satisfying the criteria of being Independent and Comprehensive.

**Checklist and Hints Extraction**   Based on the extracted questions, golden answers, and the full paper content, annotators further derive and organize hints and checklists. For hints, there are three types: background hints compiled from the introduction section of the paper, definition hints derived from core formulas and definitions in the paper, and methodology hints summarized from the main methodology section. These hints represent critical prompt information from the paper. For the checklist, annotators distill key scoring points from the golden answer, ensuring these points are verifiable and independent.

## 3.3 VALIDATION PROCESS

To ensure that each question in the benchmark strictly adheres to the design principles and expectations, and to address the issues encountered in the annotation process, we implement a multi-stage data validation pipeline. Only after successfully passing through all filtering stages and the final iterative validation loop will a task be included in the final benchmark. The validation process guideline is shown in Figure 10.

**Data Screening Principles**   The ACADREASON benchmark is built upon 50 high-level theoretical papers as targeted papers, which are selected by a panel of 10 experts specializing in five distinct fields: computer science, economics, law, mathematics, and philosophy. Annotation is performed by experts with a master's degree or higher, or those pursuing a Ph.D. or master's at leading universities, Papers are chosen according to three criteria:  1) publication in top-tier journals or conferences within their respective domains; 2) publication between 2023 and 2025; 3) purely theoretical content, excluding empirical research, reviews, and supplementary materials. These Screening principles ensure the difficulty and quality of ACADREASON. In Table 3, we present the sources of the 20 representative papers.

**Question Answerability Verification**   Since the ACADREASON benchmark requires models to conduct detailed research and demonstration, to prevent questions from being answered too broadly or evaluated ineffectively, we implement Question Answerability Verification. For each annotated question, it is assigned to three domain experts for quality inspection, the experts evaluate the questions based on three principles: clear boundaries of the question, completeness of information elements, and compliance with domain-specific logic. Only questions that meet all these criteria are retained.

## 3.4   EVALUATION METHOD AND METRICS

**Evaluation Prompt**   Previous work (Yue et al., 2024; Zhang et al., 2024a; Zhu et al., 2024; Rein et al., 2023; Ma et al., 2025) often use exact match as the evaluation metric. To provide a comprehensive evaluation framework, we select GPT-5-mini as the judge model, validated through an inter-annotator agreement (IAA) study with three independent domain experts achieving Cohen's $\kappa = 0.861$, and a multi-judge comparison showing GPT-5-mini attains 89.55% accuracy on Checklist Score against human consensus, outperforming alternatives including GPT-5, and DeepSeek-R1 in cost-efficiency (details in Appendix E.3). Given a question, the golden answer, and the corresponding checklist, the judge model evaluates the candidate's response in two aspects: (i) exact correspondence to the golden answer (1 if all required information is present and non-contradictory; 0 otherwise); (ii) independent satisfaction of each checklist item (1 if fully satisfied; 0 for partial, missing, or conflicting content). The prompt can be found in Appendix B.

**Evaluation Metrics**   We use the following two metrics as our evaluation criteria: the pass rate, measuring exact agreement with the golden answer, and the checklist score, capturing the proportion of satisfied checklist items.

- Pass Rate ($R_p$): Probability of full match with standard answers.
    - Scoring: $s_q \in \{0, 1\}$ per question.
    - Total: $R_p = \frac{\sum_{q=1}^{50} s_q}{50} \times 100$ (max =100).
- Checklist Score ($R_j$): Probability of meeting checklist criteria.
    - Scoring: $c_{q,i} \in \{0, 1\}$ per checklist item.
    - Total: $R_j = \frac{\sum_{q=1}^{50} \sum_{i=1}^{5} c_{q,i}}{250} \times 100$ (max = 100).

# 4   EXPERIMENTS

## 4.1   EXPERIMENTS SETTINGS

To comprehensively evaluate the performance of the ACADREASON benchmark, we conduct experiments from the following four perspectives: the performance of mainstream advanced reasoning models and general models on the benchmark, the performance of leading agent frameworks on the benchmark, the model performance with critical hint prompts, and a detailed analysis of failure cases. For mainstream general models and reasoning models, we directly require the models to answer the corresponding questions. For agentic frameworks, we maintain their basic tool configurations.

To further analyze the models' mastery of knowledge across different dimensions, we design detailed ablation experiments to evaluate three distinct types of hints. Finally, we also provide an analysis of

the failure reasons for current advanced models and agents, along with potential directions for future development.

**General Model & Reasoning Model** For general models and reasoning models, the acareason benchmark focuses on evaluating their knowledge reserves and reasoning capabilities. We select general models such as GPT-oss (OpenAI, 2025), GPT-4.1 (OpenAI, 2024a), GPT-5 (openai, 2025a), DeepSeek-V3 (Liu et al., 2024), DeepSeek-V3.1 (DeepSeek-AI, 2024)and Claude-Sonnet-4 (anthropic, 2025), as well as powerful reasoning models including Qwen3 (Yang et al., 2025), DeepSeek-R1 (Guo et al., 2025a), Kimi-k2 (Kimi Team et al., 2025), Gemini-2.5-Pro (Comanici et al., 2025), and o3 (OpenAI, 2024b) as our baseline models.

**Agent Framework& Agent Model** Compared to LLMs, the agent can actively gather necessary information using tools like web search and database queries, giving it enhanced retrieval capabilities. We select current state-of-the-art OAgents (GPT-5 as basic model) (Zhu et al., 2025a), Gemini-2.5-Pro-DeepResearch (google, 2025), and o3-DeepResearch (openai, 2025b) as our agent framework baselines, and Tongyi DeepResearch (Team, 2025b), AFM (Zhang et al., 2024b), MiroThinker (Team, 2025a), WebDancer (Wu et al., 2025) and WebThinker (Li et al., 2025b) as our Agent baseline.

**Ablation Experiment with Hints** To provide a more comprehensive experimental analysis and insights, we conduct an ablation study to systematically investigate the effectiveness of the multi-hint mechanism. The hints, meticulously curated by hand, encapsulate high-quality background information, methodologies, and key definitions extracted from relevant research. In this experiment, we compare baseline models without hints against ablated models integrated with hints, evaluating their performance across GPT-5, GPT-4.1, o3, and etc.

**Detailed Failure Case Analysis** The ACADREASON benchmark assesses the graduate-level reasoning abilities of LLMs, which typically require models to engage in deep thinking and generate multi-step reasoning chains. To thoroughly investigate the multi-step reasoning process and analyze failure patterns, we conduct a detailed Failure Case Analysis. We select representative models GPT-5 and OAgents, presenting their reasoning chains and logic pathways.

## 4.2 MAIN EXPERIMENT RESULT

As shown in Table 1, we present the results of over 10 LLMs and Agents. In terms of pass rate, **even the most powerful models on the market exhibit subpar performance.** For example, the latest and most powerful model, GPT-5, achieved only a 16 pass rate and a 40.6 checklist score. Most general models scored below 10 points in total. It is worth mentioning that powerful models such as GPT-4.1 and Claude-Sonnet-4 receive a score of 0, indicating that ACADREASON is highly challenging.

Compared to the stringent pass rate score, the checklist score effectively assesses how well models meet the established criteria and provides a more detailed evaluation framework. We investigate the variation in checklist scores across different academic disciplines. As shown in Figure 2, the results indicate that Computer Science and Economics exhibit relatively lower score distributions, while Law and Philosophy demonstrate higher scores. This suggests that CS and Econ present greater challenges in the ACADREASON benchmark.

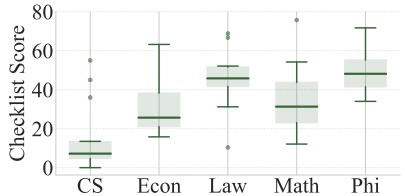

Figure 2: General performance on different domains in Checklist Score

**When comparing general models and reasoning models, the latter generally exhibit superior and more balanced performance.** We compare general models and reasoning models from the same series. For example, compared to DeepSeek-V3 (2.0/15.9), DeepSeek-R1 achieves a higher score (2.0/23.8). Similarly, o3 also outperforms GPT-4.1, demonstrating that reasoning models exhibit stronger performance within their respective series. ACADREASON focuses more on assessing the reasoning capabilities of models.

**Within the same model families, newer versions consistently outperform their older counterparts.** For example, GPT-5 outperforms GPT-4.1 in both pass rate and checklist score across multiple subjects. GPT-5 achieves an overall score of 16 and 40.5 for the pass rate and checklist score,

Table 1: Performance of various Models and Agents on ACADREASON benchmark. Each entry shows Pass Rate $R_p$ on the left and Checklist Score $R_j$ on the right. Note that the best results are in bold.

| Model | Overall | CS | Econ | Law | Math | Phi |
|---|---|---|---|---|---|---|
| *General Model* | | | | | | |
| GPT-5 | 16/40.5 | 0/13.5 | 20/46.1 | 40/52.1 | 0/51.4 | 20/56.6 |
| GPT-OSS | 4/32.2 | 0/12.6 | 0/34.2 | 10/41.7 | 10/38.3 | 0/49.1 |
| DeepSeek-V3.1 | 2/24.8 | 0/9.0 | 0/27.6 | 10/45.8 | 0/22.4 | 0/39.6 |
| DeepSeek-V3 | 2/15.9 | 0/5.4 | 10/15.8 | 0/10.4 | 0/20.6 | 0/34.0 |
| Claude-Sonnet-4 | 0/24.7 | 0/4.5 | 0/23.7 | 0/33.3 | 0/29.5 | 0/47.2 |
| GPT-4.1 | 0/21.0 | 0/0.0 | 0/18.4 | 0/31.2 | 0/31.8 | 0/37.7 |
| *Reasoning Model* | | | | | | |
| Qwen3 | 6/20.3 | 0/6.3 | 0/21.1 | 20/45.8 | 0/12.1 | 10/41.5 |
| Kimi-K2 | 6/20.3 | 0/6.3 | 0/21.1 | 20/45.8 | 0/12.1 | 10/41.5 |
| o3 | 4/33.4 | 0/8.1 | 0/38.2 | 10/50.0 | 0/40.2 | 10/50.9 |
| DeepSeek-R1 | 2/23.8 | 0/0.0 | 0/22.4 | 0/41.7 | 0/30.8 | 10/45.3 |
| Gemini-2.5-Pro | 2/22.3 | 0/2.7 | 0/15.8 | 0/41.7 | 0/25.2 | 10/49.1 |
| *Agent* | | | | | | |
| OAgents | **34/65.1** | 30/**55.0** | 30/**63.2** | 50/**68.8** | 50/**75.7** | 10/64.2 |
| Gemini-2.5-Pro-Deepresearch | 28/53.4 | **40**/45.0 | 20/56.6 | 40/66.7 | 10/44.9 | **30**/**71.7** |
| Tongyi DeepResearch | 20/30.9 | 0/5.4 | 10/34.2 | 60/62.5 | 0/32.7 | 30/47.2 |
| o3-Deepresearch | 14/47.1 | 20/36.0 | 0/38.2 | 30/52.1 | 0/54.2 | 20/64.2 |
| AFM | 14/40.5 | 10/46.5 | 0/15.8 | 40/58.3 | 10/32.7 | 10/62.3 |
| WebThinker | 8/36.4 | 22/50.0 | 0/18.4 | 10/54.2 | 0/19.4 | 11/51.1 |
| MiroThinker | 0/26.5 | 0/26.3 | 0/10.5 | 0/25.6 | 0/29.0 | 0/45.3 |
| WebDancer | 0/16.4 | 0/14.0 | 0/6.6 | 0/18.8 | 0/15.0 | 0/35.8 |

respectively, while GPT-4.1 only manages 0 and 21.0. Similarly, DeepSeek-V3.1 shows notable improvements over DeepSeek-V3, with overall scores of 2.0/24.8 and 2.0/15.9. These comparisons clearly demonstrate the positive impact of model updates and iterations on performance enhancement, newer models generally have enhanced knowledge and reasoning ability.

**Agent frameworks outperform LLMs.** OAgents achieves the best overall results among all evaluated models, with an overall pass rate of 34.0 and a checklist score of 65.1, which consistently outperform both general and reasoning models across most domains, achieving top scores in Econ, Law, Math. This is because ACADREASON contains the most challenging knowledge sections currently available as the evaluation set, which places extremely high demands on both reasoning ability and knowledge mastery. For LLMs, even though they possess strong reasoning capabilities, they lack cutting-edge academic knowledge reserve. In contrast, the agent framework can compensate for knowledge gaps through autonomous information retrieval. To verify that agent performance reflects genuine reasoning rather than source-paper retrieval, we conduct URL-masking experiments showing negligible score differences when original paper URLs are blocked (see Appendix E.1) However, the significant gap from the full score of 100 indicates that there is still room for improvement in the academic research tasks.

## 4.3 ABLATION STUDY

**The multi-hint mechanism effectively bolsters the reasoning capabilities of large language models by supplying critical contextual.** As shown in Table 2, the model's performance on ACADREASON benchmark significantly improves when hints are provided, reaching the highest score when all hints are given. Taking GPT-5 as an example, without hints, the model only achieves a score of (16.0/40.5). However, with all hints, it attains a score of (40.0/67.8), surpassing the current state-of-the-art agent framework, OAgents.

Table 2: Ablation experiment results across different hint settings. Each entry shows Pass Rate $R_p$ on the left and Checklist Score $R_j$ on the right. Note that best results are in bold.

| Models | No Hint | background | definition | methodology | ALL Hints |
|---|---|---|---|---|---|
| *General Model* | | | | | |
| GPT-5 | **16/40.5** | **16/42.5** | **24/50.9** | **34/64.3** | **40/67.8** |
| GPT-OSS | 4/32.2 | 14/40.5 | 10/42.3 | 16/52.2 | 22/58.5 |
| DeepSeek-V3.1 | 2/24.8 | 2/30.9 | 8/37.2 | 12/45.3 | 20/54.7 |
| DeepSeek-V3 | 2/15.9 | 4/25.1 | 4/26.1 | 4/38.5 | 6/44.1 |
| GPT-4.1 | 0/21.0 | 2/26.3 | 0/29.9 | 8/42.8 | 20/51.6 |
| Claude-Sonnet-4 | 0/24.7 | 2/24.6 | 2/30.6 | 14/40.8 | 11.3/49.3 |
| *Reasoning Model* | | | | | |
| Qwen3 | 6/30.4 | 14/35.7 | 10/40.5 | 20/49.1 | 22/52.7 |
| Kimi-K2 | 6/20.3 | 2/32.9 | 10/36.5 | 16/46.8 | 16/51.6 |
| o3 | 4/33.4 | 12/38.0 | 10/48.9 | 28/56.2 | 26/60.8 |
| DeepSeek-R1 | 2/23.8 | 4/30.6 | 6/35.7 | 8/45.3 | 20/50.4 |

**Different hint types provide varying benefits, with methodology hints yielding the most significant gains.** We further compare the impact of different hint types on models. As shown in the Figure 3a, we calculate the absolute gain in model accuracy for each hint type. We find that for the vast majority of models, methodology hints provide the highest gain, while background hints provide the smallest relative gain. This suggests that in ACADREASON benchmark, the focus is more on testing a model's mastery of deep methods, rather than its ability to process simple, easily accessible background information.

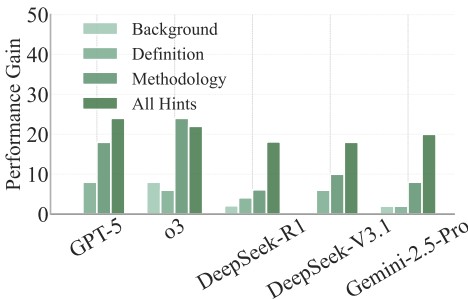

a The performance gain of various models across different hint types.

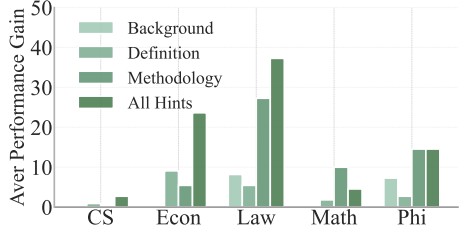

b The average performance gain across different hint types and disciplinary categories.

Figure 3: Ablation study results. (a) shows the performance gain per model, while (b) presents the average gain across disciplines.

**The benefits of different types of hints vary across different disciplines.** As shown in Figure 3b, we present the impact of different types of hints across various academic disciplines. We calculate the average improvement for all models, with additional results available in Appendix E.2. The experimental results indicate that compared to humanities subjects (Econ, Law, Phi), STEM subjects (CS, Math) achieve less improvement. This suggests that humanities disciplines place greater emphasis on the acquisition of external knowledge, while STEM fields require deeper reasoning. Furthermore, each discipline exhibits distinct focuses. For Law and Phi, hints related to methodology and background information are more important, whereas for Econ, definitions are more emphasized. This reflects the unique characteristics of different academic domains and demonstrates the comprehensiveness of the ACADREASON evaluation.

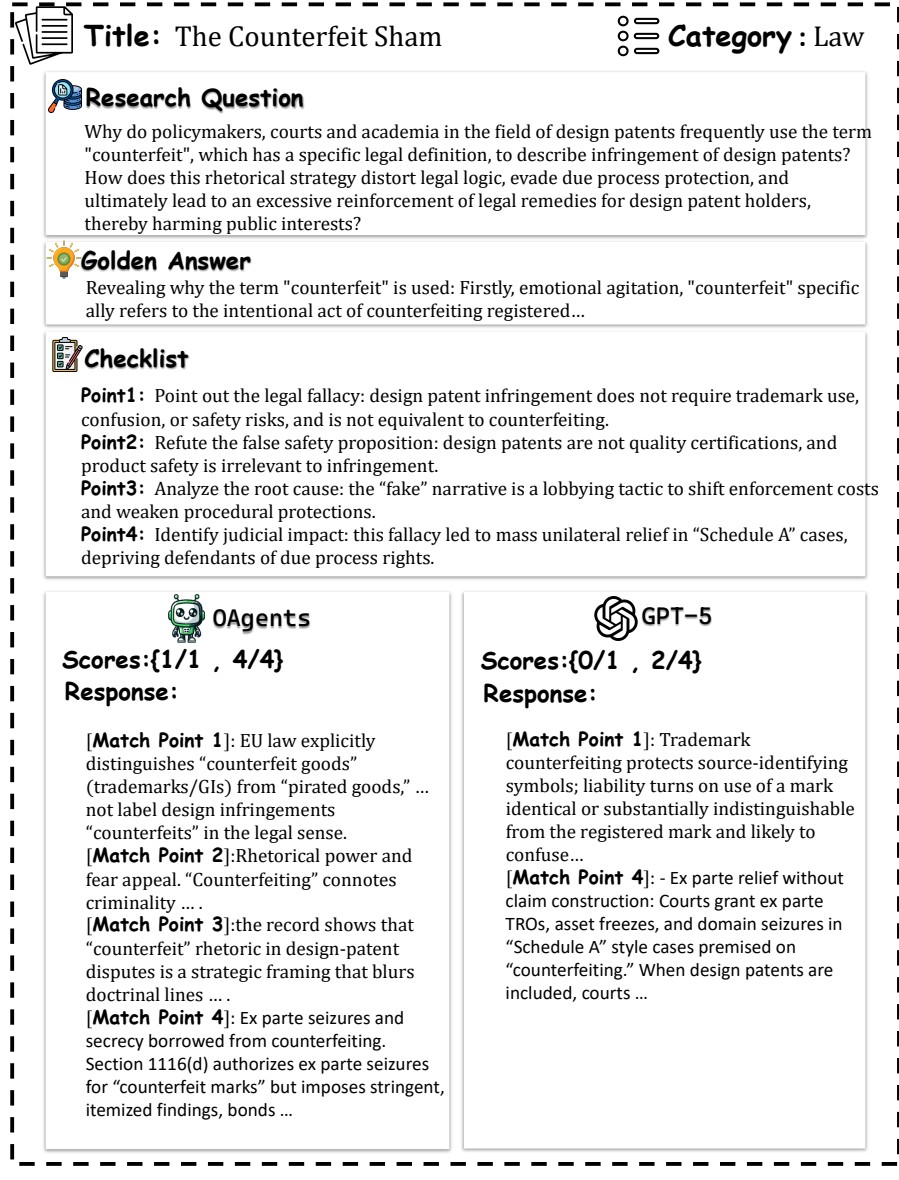

Figure 4: Side-by-side comparison of OAgents and GPT-5 on the legal reasoning task.

## 5 CASE STUDY

To provide a comparison of the leading technical paradigms, we conduct a case study featuring the top-scoring agent, OAgents (Zhu et al., 2025a), and the top-scoring single model, GPT-5 (openai, 2025a). We select a representative case from the ACADREASON benchmark where models are required to analyze the misuse of the term "counterfeit" in design patent law, as shown in Figure 4. We evaluate the models' responses against a checklist of four required actions: to point out the legal fallacy, refute the false safety proposition, analyze the root cause, and identify the judicial impact. The comparison reveals a difference: OAgents successfully address all four points for a perfect score, while GPT-5 only addresses two. OAgents provide a complete analysis, covering all required dimensions, whereas GPT-5 only succeeds in identifying the direct legal fallacy and the judicial impacts (Points 1 and 4).

The evaluation data indicates this performance gap is due to a difference in reasoning depth, not a simple lack of knowledge. GPT-5's failure on the "false safety proposition" (Point 2) stemmed from an

inability to move beyond a surface-level association of "counterfeit" with "consumer harm". It did not perform the deeper reasoning required to explicitly refute the narrative by stating that design patents are not quality certifications. Similarly, for the "root cause" (Point 3), GPT-5 identified a general "procedural leverage" but failed to synthesize this with political and economic context to identify the specific "coordinated lobbying strategy" required by the checklist. This case demonstrates that while a top-tier single model can handle direct legal analysis, the agentic framework of OAgents enables a higher-order, critical synthesis necessary to deconstruct the underlying rhetorical and political motives of a complex issue. We also provide a failure attribution analysis for Claude-Sonnet-4 in **??**.

## 6 CONCLUSION

In this work, we introduce the ACADREASON benchmark, which comprehensively evaluates the ability of LLMs and agents to acquire and reason over advanced knowledge. The ACADREASON benchmark includes 50 evaluation items across five domains, providing a comprehensive assessment of models' research capabilities. Our experimental results show that even the most advanced model, GPT-5, achieves only 16.0 points, while an advanced agent framework scores 34.0 points. These results demonstrate the difficulty and challenging nature of ACADREASON, indicating that current models still have considerable room for improvement. By releasing the entire annotated data and preliminary benchmarking results, we aim to empower the research community to better evaluate and enhance LLMs' reasoning capabilities. Our approach represents a significant step towards diversifying LLM reasoning benchmarks and utilizing the vast potential of academic research artifacts in advancing LLM research.

## 7 ETHICS STATEMENT

In the development of the ACADREASON benchmark, we have rigorously considered several ethical aspects to ensure the responsible construction and deployment of this resource.

**Data Sourcing and Intellectual Property.** All academic papers used in this benchmark are sourced from publicly available, top-tier journals and conferences. We strictly adhere to copyright laws and fair use principles for academic research. Our usage is limited to extracting research questions and creating derived reasoning tasks, without reproducing substantial copyrighted content. Each benchmark item is transformed into a novel reasoning task through significant intellectual effort, and is intended solely for academic research purposes, specifically for evaluating and advancing reasoning capabilities in AI systems.

**Expert Involvement and Compensation.** The curation process involved domain experts in selecting papers and formulating research questions. All experts were fairly compensated for their time and expertise according to academic standards, and their contributions are properly acknowledged.

## 8 REPRODUCIBILITY STATEMENT

To ensure the reproducibility of our research and facilitate future work, we have open-sourced all code and data. The project resources are available at: https://github.com/OPPO-PersonalAI/Acadreason-benchmark.

Furthermore, in the Experiment section and corresponding appendices of the paper, we provide detailed descriptions of the experimental settings and the full prompts used for reasoning and evaluation, which will fully support the replication of this study.

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

## A  DATA STATISTICS

The rigorous curation pipeline culminates in the final ACADREASON benchmark, which comprises 50 high-reasoning academic questions. In Figure 5, we present the category distribution of the dataset. Each category in the ACADREASON benchmark includes 5 samples.

Table 3 presents 20 representative papers included in the ACADREASON benchmark. All papers were selected from publicly available top-tier journals or conferences, a curation strategy that ensures the academic rigor and quality of the benchmark dataset originate from its source.

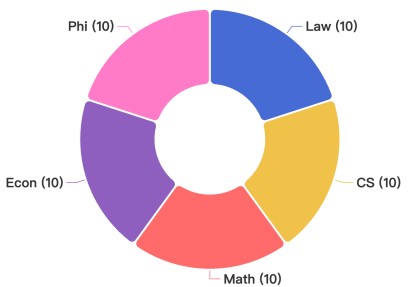

Figure 5: Category Distribution

Table 3: Representative List of 20 Papers in ACADREASON benchmark.

| Paper | Category | Source |
| --- | --- | --- |
| Reliability and Latency Analysis for Wireless Communication Systems with a Secret-Key Budget | Math | IEEE Transactions on Communications, 2024, 72(2): 1033–1044 |
| On the Popov–Belevitch–Hautus tests for functional observability and output controllability | Math | Automatica, 2025, 174(1): 112122 |
| Algebraic Geometry codes in the sum-rank metric | Math | IEEE Transactions on Information Theory, 2024, 70(5): 3345–3356 |
| Variance Decay Property for Filter Stability | Math | IEEE Transactions on Automatic Control, 2024, online first |
| Once more, without feeling | Philosophy | Philosophy & Phenomenological Research, 2025, 111(1): 343–365 |
| Patchwork ethnography | Philosophy | American Ethnologist, 2024, 51(1): 131–139 |
| Pig-feast democracy… in West Papua | Philosophy | American Ethnologist, 2024, 51(2): 193–206 |
| Moral Understanding Between You and Me | Philosophy | Philosophy & Public Affairs, 2024, 52(3): 327–357 |
| Women who pay their own brideprice… | Philosophy | Journal of the Royal Anthropological Institute, 2025, 31(2): 493–512 |
| A Lower Bound for Light Spanners in General Graphs | Computer Science | Proceedings of SODA 2025: 4327–4337 |
| Tight Streaming Lower Bounds for Deterministic Approximate Counting | Computer Science | Proceedings of SODA 2025 (Best Student Paper) |
| A Refutation of the Pach–Tardos Conjecture for 0-1 Matrices | Computer Science | Proceedings of SODA 2025 |
| Universal Perfect Samplers for Incremental Streams | Computer Science | Proceedings of SODA 2025: 3409–3429 |
| Quasi-Monte Carlo Beyond Hardy-Krause | Computer Science | Proceedings of SODA 2025: 2051–2075 |
| Waste, Property, and Useless Things | Law | Harvard Law Review, 2025, Vol. 138 (accepted) |
| The Law and Lawlessness of U.S. Immigration Detention | Law | Harvard Law Review, 2025, 138(5): 1186– |
| Human Rights Obligations in Maritime Search and Rescue | Law | International & Comparative Law Quarterly, 2025, 74(1): 33–60 |
| State Immunity from Non-Judicial Measures of Constraint | Law | International & Comparative Law Quarterly, 2025, 74(1): 179–204 |
| Informational Black Holes in Financial Markets | Economy | Journal of Finance, 2023, 78(6): 3099–3140 |
| A Theory of Dynamic Inflation Targets | Economy | American Economic Review, 2025, 115(2): 448–490 |

## B    PROMPT FOR INFER AND EVALUATION

The prompts we used for ACADREASON benchmark are shown below.

---

**PROMPT FOR INFER**

Please answer the following question:

Question: {query}

Provide a precise and detailed response.

---

Figure 6: Prompt for infer

---

**PROMPT FOR EVAL**

Task: Judge the following attempt answer to an academic question based on the provided question, checklist criteria, and golden answer reference.

Judgement Criteria
Aspect 1: Answer Correspondence
Judge if the answer corresponds to the golden answer:
– 1 point: The answer contains all the information of the golden answer
– 0 point: The answer completely fails to meet or only partially meets the key information required by the golden answer, or if there are contradictions
Aspect 2: Checklist Requirements
For every item on the checklist, judge independently whether the answer meets the requirement. The hints are provided to help you judge:
– 1 point: The reasoning and answer meet the requirement
– 0 point: The reasoning and answer do not meet the requirement, or only partially meet the requirement

Data Information
Inputs
– Question: {query}
– Checklist: {checklist
Answer to judge
– Attempted Answer: {response}
Golden Output
– Golden Answer: {golden_answer}

Output Format
Please respond strictly in the JSON format provided below. Note that the number of items in the checklist should be equal to the number of items in the justifications and scores for aspect 2. The number of checklist items can vary.

Example Output
```
{{
"aspect_1_analysis": "Give the reason for how to score the aspect 1",
"aspect_1_score": 0,
"aspect_2_analysis_1": "Give the reason for how to score the first item in the checklist in aspect 2",
"aspect_2_score_1": 0,
"aspect_2_analysis_2": "Give the reason for how to score the second item in the checklist in aspect 2",
"aspect_2_score_2": 0,
...
}}
```

---

Figure 7: Prompt for eval

## C  LIMITATIONS

The current benchmark comprises 50 expert-curated problems, a deliberate design choice given the substantial computational cost of deep research evaluation—where a single query often necessitates tens to hundreds of reasoning steps—consistent with similar benchmarks in this domain (e.g., XBench-deepsearch: 100, PaperBench: 20). As peer-reviewed publications continuously emerge across all academic domains, ACADREASON can be incrementally expanded while maintaining the annotation rigor essential for research-level evaluation. We plan to explore semi-automatic scaling approaches leveraging LLM-assisted pipelines to facilitate sustainable growth of the benchmark.

## D  LLM USAGE

Large language models (LLMs) are used in this work exclusively for text polishing and language refinement during the writing process. Specifically, LLMs assist in improving the fluency, clarity, and conciseness of the writing.
LLMs are not used for any aspects of experimental design, methodological development or scientific interpretation. All scientific contributions and innovations presented in this work are entirely human-originated.

## E  MORE EXPERIMENT RESULT

### E.1  INFORMATION LEAKAGE DETECTION EXPERIMENT

To address concerns about potential data contamination through web search capabilities, we conducted a controlled experiment by masking URLs of the original source papers. Specifically, we blacklisted all URLs containing the original paper content, preventing agents from directly accessing them during evaluation.

We evaluated two representative agent systems—OAgents and Tongyi-DeepResearch—under both masked and unmasked conditions. Table 4 presents the results.

Table 4: Performance Comparison with URL Masking

| Agent | Pass Rate (%) | Checklist Score (%) |
|---|---|---|
| OAgents (w/o mask) | 34 | 65.1 |
| OAgents (w/ mask) | 32 | 65.8 |
| Tongyi-DeepResearch (w/o mask) | 16 | 49.2 |
| Tongyi-DeepResearch (w/ mask) | 16 | 47.0 |

The results show minimal performance differences between masked and unmasked settings for both agents (differences within 2 percentage points). This robustly demonstrates that direct access to source papers does not significantly inflate agent performance, confirming that ACADREASON primarily tests reasoning capabilities rather than information retrieval. This finding can be attributed to our careful benchmark design: each research question was deliberately crafted to be highly autonomous and independent from the source paper content, requiring substantial reasoning even when the original paper is accessible.

### E.2  DETAILED TABLE ABOUT HINTS ABLATION EXPERIMENTS

Table 5: Performance of various Models on ACADREASON benchmark, providing with background hint. Each entry shows Pass Rate $R_p$ on the left and Checklist Score $R_j$ on the right. Note that best results are in bold.

| Model | Overall | CS | Econ | Law | Math | Phi |
|---|---|---|---|---|---|---|
| *General Model* | | | | | | |
| GPT-5 | **16.0/42.5** | 0.0/8.1 | 0.0/**47.4** | **50.0**/60.4 | 0.0/**49.5** | **30.0/77.4** |
| GPT-OSS | 14.0/40.5 | 0.0/**19.8** | 0.0/32.9 | 40.0/56.2 | **10.0**/42.1 | 20.0/**77.4** |
| DeepSeek-V3 | 4.0/25.1 | 0.0/2.7 | 0.0/17.1 | 10.0/47.9 | 0.0/28.0 | 10.0/56.6 |
| GPT-4.1 | 2.0/26.3 | 0.0/2.7 | 0.0/21.1 | 0.0/41.7 | 0.0/30.8 | 10.0/60.4 |
| Claude-Sonnet-4 | 2.0/24.6 | 0.0/0.0 | 0.0/17.1 | 10.0/45.8 | 0.0/29.0 | 0.0/58.5 |
| DeepSeek-V3.1 | 2.0/30.9 | 0.0/13.5 | 0.0/19.7 | 0.0/39.6 | 0.0/33.6 | 10.0/69.8 |
| *Reasoning Model* | | | | | | |
| Qwen3 | 14.0/35.7 | 0.0/6.3 | **10.0**/34.2 | 40.0/**68.8** | 0.0/31.8 | 20.0/77.4 |
| o3 | 12.0/38.0 | 0.0/7.2 | **10.0**/31.6 | 20.0/56.2 | 0.0/46.7 | **30.0**/77.4 |
| DeepSeek-R1 | 4.0/30.6 | 0.0/5.4 | 0.0/38.2 | 10.0/41.7 | 0.0/29.9 | 10.0/64.2 |
| Gemini-2.5-Pro | 4.0/26.6 | 0.0/6.3 | 0.0/23.7 | 10.0/50.0 | 0.0/20.6 | 10.0/64.2 |
| Kimi-k2 | 2.0/32.9 | 0.0/8.1 | 0.0/26.3 | 0.0/50.0 | 0.0/35.5 | 10.0/73.6 |

Table 6: Performance of various Models on ACADREASON benchmark, providing with Definition Hint. Each entry shows Pass Rate $R_p$ on the left and Checklist Score $R_j$ on the right. Note that best results are in bold.

| Model | Overall | CS | Econ | Law | Math | Phi |
|---|---|---|---|---|---|---|
| *General Model* | | | | | | |
| GPT-5 | **24.0/50.9** | 0.0/18.9 | **40.0/72.4** | **60.0/64.6** | 10.0/**55.1** | 10.0/**66.0** |
| GPT-OSS | 10.0/42.3 | 0.0/17.1 | 10.0/59.2 | 20.0/62.5 | 0.0/37.4 | **20.0**/62.3 |
| DeepSeek-V3.1 | 8.0/37.2 | 0.0/10.8 | 10.0/51.3 | 10.0/52.1 | **20.0**/41.1 | 0.0/50.9 |
| DeepSeek-V3 | 4.0/26.1 | 0.0/3.6 | 10.0/34.2 | 10.0/43.8 | 0.0/24.3 | 0.0/49.1 |
| Claude-Sonnet-4 | 2.0/30.6 | 0.0/2.7 | 0.0/38.2 | 10.0/47.9 | 0.0/35.5 | 0.0/52.8 |
| GPT-4.1 | 0.0/29.9 | 0.0/8.1 | 0.0/42.1 | 0.0/41.7 | 0.0/29.9 | 0.0/47.2 |
| *Reasoning Model* | | | | | | |
| o3 | 10.0/48.9 | **10.0**/35.1 | 0.0/61.8 | 20.0/56.2 | 0.0/42.1 | **20.0/66.0** |
| Qwen3 | 10.0/40.5 | 0.0/16.2 | 20.0/57.9 | 10.0/60.4 | 0.0/33.6 | **20.0**/62.3 |
| Kimi-K2 | 10.0/36.5 | 0.0/20.7 | 10.0/44.7 | 20.0/54.2 | 0.0/27.1 | **20.0**/60.4 |
| DeepSeek-R1 | 6.0/35.7 | 0.0/4.5 | 20.0/63.2 | 0.0/43.8 | 0.0/33.6 | 10.0/58.5 |
| Gemini-2.5-Pro | 4.0/38.5 | 0.0/8.1 | 10.0/63.2 | 0.0/41.7 | 0.0/43.0 | 10.0/54.7 |

Table 7: Performance of various Models on ACADREASON benchmark, providing with Methodology Hints. Each entry shows Pass Rate $R_p$ on the left and Checklist Score $R_j$ on the right. Note that the best results are in bold.

| Model | Overall | CS | Econ | Law | Math | Phi |
|---|---|---|---|---|---|---|
| *General Model* | | | | | | |
| GPT-5 | **34.0/64.3** | 0.0/**37.8** | 20.0/69.7 | **70.0/75.0** | **40.0/70.1** | **40.0/90.6** |
| GPT-OSS | 16.0/52.2 | 0.0/27.0 | 0.0/60.5 | 30.0/54.2 | 20.0/57.0 | 30.0/81.1 |
| Claude-Sonnet-4 | 14.0/40.8 | 0.0/13.5 | 0.0/34.2 | 50.0/62.5 | 0.0/43.0 | 20.0/83.0 |
| DeepSeek-V3.1 | 12.0/45.3 | 0.0/19.8 | 20.0/47.4 | 20.0/62.5 | 10.0/46.7 | 10.0/77.4 |
| GPT-4.1 | 8.0/42.8 | 0.0/18.9 | 0.0/43.4 | 30.0/56.2 | 10.0/46.7 | 0.0/71.7 |
| DeepSeek-V3 | 4.0/38.5 | 0.0/14.4 | 0.0/32.9 | 10.0/58.3 | 0.0/43.0 | 10.0/69.8 |
| *Reasoning Model* | | | | | | |
| o3 | 28.0/56.2 | 0.0/31.5 | **40.0/73.7** | 50.0/58.3 | 10.0/57.0 | **40.0**/79.2 |
| Qwen3 | 20.0/49.1 | 0.0/18.9 | 10.0/42.1 | 60.0/70.8 | 10.0/56.1 | 20.0/88.7 |
| Kimi-k2 | 16.0/46.8 | 0.0/22.5 | 0.0/53.9 | 40.0/64.6 | 10.0/39.3 | 30.0/86.8 |
| Gemini-2.5-Pro | 10.0/48.6 | 0.0/25.2 | 0.0/50.0 | 20.0/56.2 | 10.0/48.6 | 20.0/88.7 |
| DeepSeek-R1 | 8.0/45.3 | 0.0/16.2 | 0.0/48.7 | 20.0/50.0 | 0.0/48.6 | 20.0/**90.6** |

### E.3 VALIDATING THE LLM JUDGE: A HUMAN-ALIGNMENT STUDY

To ensure the validity and reliability of our automated evaluation approach, we conducted a systematic comparison between multiple LLM judges and human expert evaluations. This section details our methodology and findings that led to the selection of GPT-5 Mini as our primary judge model.

#### E.3.1 INTER-ANNOTATOR AGREEMENT EXPERIMENT

We sampled 10 instances from the full set of 50 data points to serve as a validation set. The instances' inference results, generated by OAgents (backbone GPT-5), were selected across a range from high to low scores to mitigate selection bias. We then invited 3 experts for each domain to independently judge these samples based on two metrics: *Pass Rate* and *Checklist Score*.

Table 8 presents the detailed comparison between GPT-5 Mini judge and expert human judges across different domains and samples.

Table 8: Comparison of GPT-5 Mini Judge and Human Expert Evaluations

| ID | Domain | Pass Rate | | | | Checklist Score | | | |
|----|--------|-----------|--------|--------|--------|-----------------|--------|--------|--------|
| | | GPT-5 Mini | Human1 | Human2 | Human3 | GPT-5 Mini | Human1 | Human2 | Human3 |
| 3 | Philosophy | 1/1 | 1/1 | 1/1 | 1/1 | 4/4 | 4/4 | 4/4 | 3/4 |
| 6 | Philosophy | 0/1 | 0/1 | 0/1 | 0/1 | 1/6 | 1/6 | 2/6 | 2/6 |
| 11 | Computer Science | 0/1 | 0/1 | 0/1 | 0/1 | 4/6 | 0/6 | 1/6 | 1/6 |
| 12 | Computer Science | 0/1 | 0/1 | 0/1 | 0/1 | 2/8 | 2/8 | 2/8 | 2/8 |
| 26 | Law | 1/1 | 1/1 | 1/1 | 1/1 | 6/6 | 6/6 | 6/6 | 6/6 |
| 30 | Law | 0/1 | 0/1 | 0/1 | 0/1 | 0/5 | 0/5 | 0/5 | 0/5 |
| 34 | Economics | 0/1 | 0/1 | 0/1 | 0/1 | 4/7 | 3/7 | 5/7 | 4/7 |
| 40 | Economics | 1/1 | 0/1 | 0/1 | 0/1 | 8/8 | 7/8 | 8/8 | 6/8 |
| 44 | Math | 1/1 | 0/1 | 0/1 | 0/1 | 9/9 | 5/9 | 5/9 | 5/9 |
| 48 | Math | 0/1 | 0/1 | 0/1 | 0/1 | 5/8 | 5/8 | 4/8 | 5/8 |

For each domain, we engaged three independent human annotators to label the samples. The inter-annotator agreement was measured using Cohen's $\kappa$ for each annotator pair, yielding an average $\kappa$ of 0.861 (range: 0.843–0.870). Ground truth labels were established through majority voting among the three annotators, which we then used to calculate the consistency score with a series of candidate LLM judge models.

#### E.3.2 MODEL COMPARISON AND CONCLUSION

We evaluated multiple candidate models against the established ground truth. Tables 9 and 10 present the consistency metrics for Pass Rate and Checklist Score, respectively.

Table 9: Consistency Metrics for Pass Rate Evaluation

| Model | Acc (%) | Prec (%) | Recall (%) | F1 (%) | Cost ($/1M-tokens) |
|-------|---------|----------|------------|--------|---------------------|
| Random | 50 | 0 | 0 | 0 | – |
| GPT-5 | 90 | 66.67 | 100 | 80 | 1.25/10 |
| GPT-5 mini | **90** | **66.67** | **100** | **80** | **0.25/2** |
| Claude-Sonnet-4.5 | 80 | 50 | 100 | 66.67 | 3/15 |
| DeepSeek-V3 | 70 | 40 | 100 | 57.14 | 0.28/0.42 |
| DeepSeek-R1 | 80 | 50 | 50 | 50 | 0.55/2.19 |

Table 10: Consistency Metrics for Checklist Score Evaluation

| Model | Acc (%) | Prec (%) | Recall (%) | F1 (%) | Cost ($/1M-tokens) |
|-------|---------|----------|------------|--------|---------------------|
| Random | 49.75 | 48.85 | 45.45 | 47.09 | – |
| GPT-5 | 86.57 | 92.86 | 78.79 | 85.25 | 1.25/10 |
| GPT-5 mini | **89.55** | **86.11** | **93.94** | **89.86** | **0.25/2** |
| Claude-Sonnet-4.5 | 85.07 | 89.66 | 78.79 | 83.87 | 3/15 |
| DeepSeek-V3 | 82.09 | 74.42 | 96.97 | 84.21 | 0.28/0.42 |
| DeepSeek-R1 | 85.07 | 89.66 | 78.79 | 83.87 | 0.55/2.19 |

As demonstrated by the results, GPT-5 Mini achieves high overall consistency scores across both metrics with human expert evaluations, robustly validating its strong alignment with human judgment. Notably, GPT-5 Mini reduces the evaluation cost by 80% compared to GPT-5, while also offering significantly faster inference speed.

Specifically, for Pass Rate evaluation, GPT-5 Mini achieves 90% accuracy with perfect recall (100%), matching the performance of GPT-5 while being substantially more cost-effective. For Checklist Score evaluation, GPT-5 Mini demonstrates superior performance with 89.55% accuracy and 89.86% F1 score. Considering the essential trade-off between performance, cost, and efficiency for large-scale evaluation, we ultimately selected GPT-5 Mini as our primary judge model. This combination of high performance, low cost, and fast inference makes GPT-5 Mini the optimal choice for our large-scale evaluation framework.

### E.4 COMPARISON WITH OTHER BENCHMARKS

To intuitively illustrate the differences between ACADREASON and existing benchmarks, we provide a systematic comparison with related mainstream benchmarks, including PaperBench, HLE, BrowseComp, XBench-DeepSearch, GAIA, and DeepResearchBench.

### E.4.1 QUALITATIVE ANALYSIS

These benchmarks generally cover three types of tasks: (1) **Code reproduction** (e.g., PaperBench): given a paper as input, the goal is to reproduce the corresponding repository; (2) **Search/QA** (e.g., HLE, GAIA, BrowseComp, XBench-DeepSearch): the core capability tested is information retrieval and short-form question answering; (3) **Open-ended research/report** (e.g., DeepResearchBench): given semi-open-ended questions such as "How to enhance classroom participation for students with autism?", the model performs broad research and provides a report. Table 11 summarizes the key characteristics of these benchmarks in terms of domain coverage, task type, and output format.

Table 11: Comparison of ACADREASON with Related Benchmarks

| Benchmark | Domain Numbers | Task Type | Output Format |
|---|---|---|---|
| PaperBench | 1 | Code reproduction | Repo/Code |
| HLE | 8 | Expert-level reasoning QA | Short QA |
| BrowseComp | 1 | Search-based QA | Short QA |
| XBench-DeepSearch | 1 | Search-based QA | Short QA |
| GAIA | 5 | Assistant-style QA (web, code, multimodal) | Short QA |
| DeepResearchBench | 22 | Research-style information gathering | Long-form report |
| ACADREASON (Ours) | 5 | Research-level multi-step reasoning | Long-form report |

Our benchmark is specifically designed for the "research-level long report" scenario on academic research problems. Given a specific research question, the model must summarize the status quo, perform multi-step reasoning, and provide a solution-oriented research report—mimicking the workflow of a human researcher. This task setting is currently absent in existing benchmarks.

Compared with XBench-DeepSearch and BrowseComp, we do not primarily evaluate long-chain retrieval capability itself. Instead, we focus on whether the model can complete research-level comprehensive analysis and compose a complete long-form report under the premise of having obtained relevant evidence.

Unlike HLE and GAIA, which also target academic domains but adopt a QA format focusing on retrieval correctness and short answers, our task requires the model to conduct systematic research and output a structured, long-form report.

PaperBench centers on code reproduction with repository outputs as the goal. While DeepResearch-Bench also requires report generation, it uses semi-open-ended questions from public domains (e.g., "How to enhance classroom participation for students with autism?"). In contrast, ACADREASON focuses on specific academic research problems, emphasizing problem decomposition, literature review, and solution reasoning in a manner consistent with human researchers' methodologies.

### E.4.2 QUANTITATIVE DIFFICULTY ANALYSIS

To quantitatively demonstrate ACADREASON's challenging nature, we compared the performance of state-of-the-art models and agents across multiple benchmarks. Tables 12 and 13 present performance comparisons across different evaluation frameworks.

Table 12: Cross-Benchmark Performance Comparison for Agent Systems

| Model/Agent | HLE | GAIA | BrowseComp | **ACADREASON** |
|---|---|---|---|---|
| Tongyi-DeepResearch | 32.9 | 70.9 | 43.4 | **16.0** |
| AFM | 18.0 | 55.3 | 11.1 | **14.0** |
| WebThinker | 15.8 | 48.5 | – | **8.0** |

Table 13: Model Performance Across Academic Reasoning Benchmarks

| Model | HLE (Academic) | GPQA Diamond (Scientific) | **ACADREASON** |
|---|---|---|---|
| GPT-5 | 25.32 | 84.2 | **16.0** |
| Gemini-2.5-Pro | 18.08 | 84.0 | **2.0** |
| Kimi-K2 | 75.1 | 4.7 | **6.0** |
| Qwen3 | – | 71.1 | **6.0** |
| DeepSeek-R1 | – | 71.5 | **2.0** |

The results reveal a consistent pattern: models and agents achieve substantially lower scores on ACADREASON compared to other benchmarks. For instance, Tongyi-DeepResearch scores 70.9% on GAIA but only 16.0% on ACADREASON—a 54.9 percentage point drop. Similarly, GPT-5 and Gemini-2.5-Pro achieve over 84% on GPQA Diamond yet score only 16.0% and 2.0% respectively on ACADREASON. These dramatic performance gaps suggest that ACADREASON measures distinct capabilities—specifically, the ability to conduct deep, multi-step research-level reasoning—that are not adequately captured by existing benchmarks focused on factual recall or search-based question answering.

To provide a comprehensive quantification of ACADREASON's difficulty, we computed aggregate statistics across all 19 evaluated models and agents. The remarkably low average Pass Rate (8.53%, std: 9.97%) and moderate Checklist Score (31.34%, std: 13.20%) demonstrate that ACADREASON poses substantial challenges even to frontier systems. The moderate standard deviation indicates that while the benchmark is difficult, it maintains sufficient discriminative power across different capability levels. The gap between Pass Rate and Checklist Score suggests that models can partially complete reasoning steps but struggle to produce fully correct, comprehensive solutions—a pattern consistent with the research-level nature of our tasks.

# F    ANNOTATION AND VALIDATION GUIDELINE

---

**RESEARCH QUESTIONS ANNOTATION GUIDELINE**

Your task is to extract one high-quality research question from a provided academic paper and then construct a comprehensive golden answer for it. The final question-answer pair should be self-contained, accurately reflecting the paper's core theoretical contribution, and must be solvable without requiring access to the original text. The primary goal is to create a challenging benchmark item that tests advanced reasoning.

**Research Question**

- **Clarity and Self-consistency:** Questions should have well-defined boundaries and include minimal necessary background, focusing on specific theoretical problems.
- **Alignment and Independence:** Questions must align with the paper's core contribution and be answerable without requiring access to the full text.
- **Structural Constraints:** Avoid open-ended formulations, composite questions requiring decomposition, or references to the original paper's structure.

**Golden Answer**

- **Comprehensive Coverage:** Answers should cover background, definitions, derivations/proofs, and conclusions, satisfying all checklist requirements.
- **Verifiability:** Provide key intermediate steps and essential formulas to ensure reproducibility and self-contained reasoning.
- **Content Integrity:** Maintain logical continuity without skipping critical steps, introducing external information, or violating domain-specific conventions.

Figure 8: Guideline For Research Question Annotation

---

**HINTS AND CHECKLIST ANNOTATION GUIDELINE**

Your task is to create Hints and a Checklist based on the provided Research Question, Golden Answer, and the original paper. The Hints should provide necessary but incomplete support for reasoning, while the Checklist must enable clear and objective verification of a complete answer.

**Hints Annotaion**

- **Background:** Context from the introduction and related work necessary to understand the problem.
- **Definitions:** Standardized statements of core concepts and terminology.
- **Methods:** Essential theoretical tools, methodological frameworks, and key technical tips.
- **Selection Principle:** Include only information necessary to facilitate reasoning, avoiding final conclusions or direct answers.

**Checklists Annotaion**

- **Atomicity:** Each item contains a single step or fact.
- **Decidability:** Criteria for fulfilling each item are clear and binary.
- **Independence:** Minimizing dependencies between different items.
- **Source:** Key steps and evidential facts are extracted from the Golden Answer.
- **Explicit Referencing:** Phrasing items as checks for specific statements (e.g., "Did it prove that [statement]?") instead of referencing internal labels.

Figure 9: Guideline For hints and checklist Annotation

**VALIDATION GUIDELINE**

Your task is to ensure the creation of a high-quality dataset for complex reasoning. You will be responsible for reviewing and refining data items, each consisting of a Research Question, Hints, a Checklist, and a Golden Answer.

**Data Screening**

- **Source Verification:** Confirm the academic authority and timeliness of data sources.
- **Content Qualification:** Ensure the content is purely theoretical, excluding applied and empirical materials.
- **Difficulty Assessment:** Filter for problems with high reasoning complexity and intellectual challenge.

**Question Answerability Verification** The core principle is to ensure the question itself is well-defined and answerable.

- **Clear Boundaries:** The input conditions, solution scope, and final objectives of the question must be unambiguous.
- **Complete Information:** Provide the minimal necessary background knowledge and key information points required for understanding and solving the problem.
- **Logical Compliance:** The problem statement and reasoning process must strictly adhere to the norms and theoretical framework of the respective discipline.

**Consistency Check** Conduct a systematic verification of the four core components that constitute a complete data item:

- **Overall Consistency:** The Question, Hints, Checklist, and Golden Answer must be logically self-consistent, mutually supportive, and free of contradictions.
- **Verifiability of Checklist Items:** Each item in the checklist must correspond to explicit evidence in the Golden Answer, with verification criteria that are clear and actionable.

Figure 10: Guideline For Quality Validation

## G   SPECIFIC CASE OF ACADREASON BENCHMARK

---

**PHILOSOPHY**

**Title:** Moral Understanding Between You and Me
**Category:** Philosophy
**Research Question:** Why shared moral understanding is important?
**Golden Answer:**

Moral understanding is an epistemic achievement about moral matters. To have it, your conception of a moral issue must be accurate (the objective dimension) and also "make sense to you" (the subjective dimension); your moral beliefs must be based directly in the reasons that make them true. You can know more than you understand; for example...

Shared Understanding as the Aim of Interpersonal Justification,Within the central moral practices of interpersonal justification...

Shared Understanding and the Norms of Apology,An apology should aim to reflect a shared moral understanding of the wrong...

Reasoning with the Unreasonable,A worry about the shared understanding requirement is...

**Checklist:**

1. Does the answer include content related to sharing moral understanding?
2. Does the answer include the view that the constitutive aim of interpersonal justification is shared moral understanding?
3. Does the answer include content on shared understanding in the context of interpersonal justification?
4. Does the answer include the view that an apology should aim to reflect a shared moral understanding of the wrong done to its recipient?
5. Does the answer include content on the Shared Understanding Condition of apology?

**Hints:**

1. **Background:** Much attention has been paid to moral understanding as an individual achievement, when a single agent gains insight into distinctly moral matters. But the importance of moral understanding cannot be fully explained by merely focusing on individuals' moral understanding...
2. **Definition:** understanding:The capacity to grasp the moral significance of actions, principles, or situations. It involves not only knowing moral facts or rules but also appreciating the reasons behind them, recognizing the perspectives and experiences of others, and being able to make sense of moral demands in context...
3. **Methodology:** giving an account of what it takes for you and me to share moral understanding.2.Through comparison of the Delivery Model, the moral address view, and shared moral understanding, the constitutive aim of interpersonal justification is clarified as shared moral...

---

Figure 11: The sample of Philosophy domain

---

**MATH**

**Title:** Sorting permutations using a pop stack with a bypass
**Category:** Math
**Research Question:**

How can permutations be characterized and enumerated under sorting by a pop stack equipped with a bypass operation? In particular, which forbidden patterns give necessary and sufficient criteria for sortability, how can a bijection with suitably restricted Motzkin paths be constructed so that the counting sequence is the odd-indexed Fibonacci numbers, and how can one design and analyze an algorithm to compute preimages—especially for permutations with few preimages and for principal classes—with a structural description of these sets? Furthermore, how do these results extend to several pop stacks in parallel with bypass, yielding explicit bases for the sortable permutations, rational generating functions, and connections to classical sorting algorithms, with rigorous proofs throughout?

**Golden Answer:**

Pattern Characterization and Algorithm Optimality: Permutations sortable by the pop stack with bypass (PSB) are precisely those that avoid the patterns 231 and 4213...

Enumeration via Motzkin Paths and Fibonacci Numbers: Sortable permutations can be encoded as ternary words built from the PSB operations (PUSH $= 0$, BYPASS $= 1$, POP+PUSH $= 2$)...

Preimages under PSB: Every permutation has a well-defined set of preimages under PSB. The algorithm for constructing preimages relies on decomposing a permutation by its left-to-right maxima and...

Preimages of Permutation Classes: For certain principal classes, preimages under PSB remain classes. If the basis permutation begins with its maximum ($n\alpha$) or begins with the second maximum...

**Checklist:**

1. Defines fundamental concepts: permutation $\pi$, pop stack operations (PUSH, POP, BYPASS), and the pattern avoidance framework.
2. Characterizes PSB-sortable permutations by avoidance of patterns 231 and 4213, showing necessity and sufficiency.
3. Establishes a bijection between sortable permutations and restricted Motzkin paths, proving the enumeration equals odd-indexed Fibonacci numbers.
4. Provides an algorithm for computing preimages under PSB and analyzes its correctness.
5. ...

**Hints:**

1. **Background:** 1. Sorting permutations in combinatorics | • Central research topic, studied through containers like stacks, queues, and pop stacks. | • Pattern...
2. **Definition:** 1. Permutation basics | • A permutation $\pi$ of size $n$ is a bijection from $[1, n]$ to $[1, n]$, written as $\pi = \pi_1 \cdots \pi_n$. | • Identity permutation: $\mathrm{id}_n = 12 \cdots n$. | • Sets: $S_n = $ all permutations of size $n$, $S = \bigcup S_n$. |
3. **Methodology:** 1. Sortability characterization | • Goal: determine necessary and sufficient conditions for PSB sortability....

Figure 12: The sample of Math domain

---

**LAW**

**Title:** The Dilemmas of Schrödinger's Citizenship

**Category:** Law

**Research Question:** Exploring the contradiction of whether an individual can simultaneously hold the citizenship of one or more countries and be stateless (the "Schrödinger's citizen" dilemma), analyzes its legal roots and the impact on the international human rights system.

**Golden Answer:**

> The Essence of the Theoretical Dilemma. There is a fundamental split in citizenship: The declarative nature asserts that nationality is a declaration of natural rights (such as the principle of bloodline), and requires that the effect of nationality be retroactive to birth...
>
> Reconstruction plans for the third-level systems. Level One: Amendment to international law. Amend Article 1A(2) of the Refugee Convention: A. An explanatory clause has been added to clarify that the determination of "multiple nationalities" requires meeting three conditions: ...
>
> Level Two: Reform of domestic laws. Establish a special tribunal for judicial review of nationality: A. In the initial trial stage, two types of cases are distinguished: for those involving the determination of foreign nationality...
>
> ...

**Checklist:**

1. Point out the core issue: three major problems in determining citizenship (retroactivity, foreign courts' power, and mismatch of rights with operability).
2. Conflict of laws path: prohibit foreign courts from interpreting nationality laws, require nationality review courts within the sovereign state, and ensure independent review of domestic cases.
3. Human rights path: establish graded responses to statelessness, redefine refugee standards, and change "potential nationality" to "actual administrative feasibility".
4. Institutional design: eliminate obstacles to naturalization by setting a maximum processing time and creating cross-border nationality verification centers.

**Hints:**

1. **Background:** 1.Theoretical Background | (1The declaratory-constitutive dichotomy (Ross, Austin): Legal acts are divided into declaring natural facts (such as birth) and creating new rights (such as naturalization). | (2The theory of exclusive state sovereignty over nationality (Article 1 of the Hague Convention on Nationality)...
2. **Definition:** 1.Schrödinger citizenship: A legal status where an individual is entitled to the nationality of a certain country under law, but in practice, it is not recognized by that country and is forcibly attributed by a third country. | 2.Declaratory citizenship...
3. **Methodology:** 1.Normative Analysis research method: Deconstructing the Semantic Ambiguity of the "Multiple Nationality" Clause in the Refugee Convention. | 2.Empirical research method: Citing the naturalization rate of the United Arab Emirates in 2010...

Figure 13: The sample of Law domain

---

**COMPUTER SCIENCE**

**Title:** Tight Bounds and Phase Transitions for Incremental and Dynamic Retrieval

**Category:** Computer Science

**Research Question:** Determining the optimal redundancy $R$ for retrieval data structures in the incremental and dynamic settings when the universe size is polynomial, i.e., $|\mathcal{U}| = (n)$.

**Golden Answer:**

For a polynomial universe $|\mathcal{U}| = \mathrm{poly}(n)$, the optimal redundancy $R := S - nv$ for retrieval data structures is: Incremental setting (insert-only). The optimal redundancy is:

$$R_{\mathrm{inc}} = \Theta\Big(n \;+\; n \cdot \max\{0,\; \log(\tfrac{\log n}{v})\}\Big).$$

Equivalently:

- If $v \geq c \log n$ (for a constant $c > 0$), then $R_{\mathrm{inc}} = \Theta(n)$.
- If $v = \log n / \log\log n$, then $R_{\mathrm{inc}} = \Theta(n \log\log\log n)$.
- If $v = \log^{0.99} n$, then $R_{\mathrm{inc}} = \Theta(n \log\log n)$.

These bounds are tight: there is an incremental structure with

$$S \leq nv + O(n) + O\Big(n \log\Big(\tfrac{\log n}{v}\Big)\Big)$$

and a matching lower bound

$$S \geq nv + \Omega(n) + \Omega\Big(n \log\Big(\tfrac{\log n}{v}\Big)\Big) \quad (\text{for } |\mathcal{U}| \geq n^3),$$

giving the phase transition around $v \asymp \log n$. Timewise...

**Checklist:**

1. Setup – States $|U| = \mathrm{poly}(n)$. – Defines redundancy $R := S - nv$.
2. Incremental formula – Gives $R_{\mathrm{inc}} = \Theta\big(n + n \cdot \max\{0, \log(\tfrac{\log n}{v})\}\big)$. – Mentions phase transition at $v \asymp \log n$.
3. Incremental cases – $v \geq c \log n \;\Rightarrow\; R_{\mathrm{inc}} = \Theta(n)$. – $v = \tfrac{\log n}{\log\log n} \;\Rightarrow\; R_{\mathrm{inc}} = \Theta(n \log\log\log n)$. – $v = \log^{0.99} n \;\Rightarrow\; R_{\mathrm{inc}} = \Theta(n \log\log n)$.
4. Incremental bounds – Upper bound: $S \leq nv + O(n) + O\big(n \log(\tfrac{\log n}{v})\big)$. – Lower bound: $S \geq nv + \Omega(n) + \Omega\big(n \log(\tfrac{\log n}{v})\big)$, for $|U| \geq n^3$.
5. ...

**Hints:**

1. **Background:** Retrieval data structures are designed to answer key-value queries without explicitly storing the keys...
2. **Definition:** - **Retrieval Data Structure:** A data structure that answers key-value queries without storing keys explicitly. Given a set of $n$ keys $K \subseteq \mathcal{U}$ and a $v$-bit value $f(k)$ for each key $k \in K$, it supports queries of the form:
$$\mathtt{Query}(k) = \begin{cases} f(k), & \text{if } k \in K, \\ \text{anything}, & \text{otherwise.} \end{cases}$$

Figure 14: The sample of Computer Science domain

---

**ECONOMICS**

**Title:** Informational Black Holes in Financial Markets
**Category:** Economics
**Research Question:**

Suppose a project can be either a benign type $G$ or an inferior type $B$, with investors receiving independent private signals that satisfy the strict monotone likelihood ratio property (MLRP). The project is undertaken if and only if at least one investor participates. Please address the following: In a competitive market with $N$ investors, demonstrate rigorously that a robust symmetric equilibrium is characterized by a unique participation threshold, $s_N \in (0, 1)$, where an investor $i$ participates if and only if their signal $s_i \geq s_N$. Explain the economic intuition behind the existence and uniqueness of this threshold. In the limit as $N \to \infty$, the number of participants, $\kappa$, converges to a Poisson distribution. Derive the limiting distributions for $\kappa$ conditional on project types $G$ and $B$. Furthermore, provide a complete derivation for the closed-form expression of the parameter

$$\tau = \lim_{N \to \infty} N \cdot \Pr(S_i \geq s_N \mid \theta = B),$$

expressing it in terms of the signal's top likelihood ratio $(\lambda)$, the prior probability $(\pi_0)$, and the project's break-even posterior probability $(\pi^*)$.

**Golden Answer:**

Existence and Uniqueness of the Participation Threshold $(s_N)$: A robust symmetric equilibrium is one that holds even with a small, non-zero participation cost. The equilibrium is characterized by a unique cutoff $s_N$ satisfying the marginal investor's indifference condition—i.e., the investor with signal $s_N$ is exactly indifferent between participating or not.

*Asymptotic Analysis:* As $N \to \infty$, the threshold $s_N \to 1$. The probability of any single investor participating, $\Pr(S_i \geq s_N)$, approaches zero. The number of participants, $\kappa_N$, which follows a binomial distribution, therefore converges to a Poisson distribution under these conditions.

The rate parameter $\tau$ is defined under the bad state ($\theta = B$) as:

$$\tau = \lim_{N \to \infty} N \cdot \Pr(S_i \geq s_N \mid \theta = B),$$

and can be expressed as a function of $(\lambda, \pi_0, \pi^*)$ based on the likelihood ratio and posterior thresholds.

**Checklist:**

1. Define robust equilibrium and explain how the winner's curse creates a threshold participation strategy.
2. Prove existence and uniqueness of the participation threshold $s_N$ via the marginal investor's break-even condition.
3. Define parameter $\tau$ as the limit of $N \cdot p_B$, where $p_B$ is participation probability in the bad state.
4. ...

**Hints:**

1. **Background:** This problem is rooted in the economic theory of asymmetric information, where different parties in a transaction hold unequal knowledge...
2. **Definition:** Strict Monotone Likelihood Ratio Property (MLRP): The ratio of conditional densities $f_G(s)/f_B(s)$ is strictly increasing in the signal $s$. This ensures a higher signal is unambiguously "good news." ...
3. **Methodology:** The existence of the threshold $s_N$ is proven by analyzing the zero-profit condition for the marginal investor,...

Figure 15: The sample of Economics domain

