# OpenReview forum: "ACADREASON: Exploring the Limits of Reasoning Models with Academic Research Problems"
_ICLR.cc/2026/Conference — ICLR 2026 Poster_

### Official Review · Reviewer_Gkt7 · 2025-10-29

**Soundness:** 3
**Presentation:** 3
**Contribution:** 3
**Rating:** 6
**Confidence:** 4

**Summary:**

This paper presents a new benchmark ACADREASON.

- This benchmark is designed to be a challenging, reasoning-intensive benchmark for academic domains, including CS, Law, Econ, Math and Philosophy. Each domain contains 10 questions.

- The benchmark is heavily human expert curated. The questions and answers are extracted and formulated by human experts from latest publications, along with hints and scoring checklist.

- The paper also benchmarked the performance of latest Large Reasoning Model and Tool-used agents on ACADREASON. The results show the benchmark is very challenging.

**Strengths:**

- The proposed benchmarks is a good contribution to the community. It has detailed and careful human expert annotations, which is a good effort.

- The benchmark is well-positioned, as it focuses on challenging reasoning and academic domains.

- The paper has benchmarked a wide range of latest LLMs and reasoning paradigms.

**Weaknesses:**

- The benchmark is relatively small, 50 questions in total. Although the scoring hints can give a bit more fine-grained signals, but in general the size is limited.

- It would be better to include a comparison section against other relevant benchmarks.

**Questions:**

- As one aspect of the contributions is the difficulty, would it be possible to quantify the difficulty and compare with other benchmark in a more explicit way?

---

> ### Author Response · Authors · 2025-11-25
> **Response to Reviewer Gkt7**
>
> We sincerely appreciate your thoughtful review and your recognition of ACADREASON's contributions, particularly the expert curation effort and challenging positioning. Your feedback on benchmark comparison and difficulty quantification is very valuable. We are happy to address these points with additional analysis and clarification.
>
> > **Q1:** As one aspect of the contributions is the difficulty, would it be possible to quantify the difficulty and compare with other benchmark in a more explicit way?
>
> Thank you for this important question about our benchmark's construct validity. To provide a more intuitive demonstration of the difficulty comparison between our benchmark and others, we evaluated the same models/agents across different benchmarks. As shown in the tables below, the models' scores on ACADREASON are significantly lower than those on other comparable benchmarks, demonstrating ACADREASON's challenging nature.
>
> |                     | HLE  | GAIA | BrowseComp | Ours |
> | ------------------- | ---- | ---- | ---------- | ---- |
> | Tongyi DeepResearch | 32.9 | 70.9 | 43.4       | 20   |
> | AFM                 | 18   | 55.3 | 11.1       | 14   |
> | Webthinker          | 15.8 | 48.5 | **—**      | 8    |
>
> | Model          | HLE (academic reasoning) | GPQA Diamond(Scientific knowledge) | Ours |
> | -------------- | ------------------------ | ---------------------------------- | ---- |
> | GPT-5          | 25.32                    | 84.2                               | 16   |
> | Gemini-2.5-Pro | 18.08                    | 84                                 | 2    |
> | Kimi-k2        | 75.1                     | 4.7                                | 6    |
> | Qwen3          | **—**                    | 71.1                               | 6    |
> | DeepSeek-R1    | **—**                    | 71.5                               | 2    |
>
> To further quantify the difficulty of ACADREASON, we computed aggregate statistics across all 19 evaluated models and agents. As shown in the table below, the overall average Pass Rate is only **8.53%** with a standard deviation of 9.97, and the average Checklist Score is **31.34%** with a standard deviation of **13.20.** These low average scores demonstrate that ACADREASON poses substantial challenges even to frontier systems, while the moderate standard deviation indicates the benchmark maintains discriminative power across different capability levels.
>
> | **Metric**          | **Mean** | **Std Dev** | **Variance** |
> | ------------------- | -------- | ----------- | ------------ |
> | Pass Rate (%)       | 8.53     | 9.97        | 99.49        |
> | Checklist Score (%) | 31.34    | 13.2        | 174.14       |
>
> > W1: The benchmark is relatively **small**, 50 questions in total. Although the scoring hints can give a bit more fine-grained signals, but in general the size is limited.
>
> Thank you for your concern. We acknowledge that 50 questions may appear limited, but this scale is deliberate and consistent with the nature of research-level reasoning benchmarks. As detailed in **General Response (part 1/3)**, deep research scenarios require extensive multi-step reasoning (often tens to hundreds of steps per query), which imposes substantial computational costs on both the evaluation infrastructure and the models being tested. This constraint is reflected across similar benchmarks in this domain (e.g., XBench: 100, PaperBench: 20).
>
> > W2: It would be better to include a **comparison section against other relevant benchmarks.**
>
> Thank you for this suggestion. We have now added a comprehensive comparison section in **General Response (part 3/3)** that positions ACADREASON relative to existing benchmarks such as **GAIA, PaperBench, DeepResearchBench, GPQA, and HLE.**
>
> Our benchmark uniquely focuses on the **"research-level long report" scenario** within academic research problems: given a specific research question, the model is required to act as a researcher by surveying the current state-of-the-art, performing multi-step reasoning, and presenting a comprehensive solution report. We believe this task setting closely mirrors the actual demands and workflows of real-world academic research, where researchers must synthesize existing knowledge, develop novel insights, and communicate findings comprehensively. This specific task setting remains unfilled by existing benchmarks.
>
> If there are any further questions or clarifications needed, we would be happy to address them. Thank you again for your thoughtful and constructive feedback.

---

### Official Review · Reviewer_RYgK · 2025-10-29

**Soundness:** 2
**Presentation:** 2
**Contribution:** 3
**Rating:** 6
**Confidence:** 4

**Summary:**

The paper presents ACADREASON, a new benchmark designed to evaluate academic-level reasoning of large language models (LLMs). Unlike existing reasoning benchmarks such as MMLU-Pro, GPQA, or PaperBench, which focus on either broad factual understanding or narrow scientific tasks, ACADREASON targets deep, research-oriented reasoning across five academic domains: Computer Science, Mathematics, Economics, Law, and Philosophy. The authors curate questions from over 400 research papers and design multi-step annotations. They evaluate several frontier models (GPT-5, DeepSeek-R1, Claude 3.7, Gemini 2.0, etc.) and find that all perform far below human experts, especially on methodological and conceptual reasoning. The study also analyzes the role of hints and long-form responses, showing that factual hints help more than methodological ones.

**Strengths:**

- High-quality benchmark design. The dataset is small but carefully curated. Questions are derived from genuine academic contexts rather than textbook or competition problems, giving ACADREASON a strong realism advantage. The multi-domain design broadens evaluation coverage beyond STEM, incorporating social science and philosophy.
- Transparent and rigorous annotation process. The paper clearly documents every stage: data sourcing, question formulation, verification, and evaluation. The inclusion of structured checklists and fine-grained rubrics for reasoning quality (clarity, coherence, accuracy) improves reproducibility and reliability compared with prior subjective benchmarks.
- Insightful empirical findings. The experiments reveal important trends: reasoning models still struggle on conceptual abstraction and logical grounding even when they perform well on applied math or coding tasks. The “hint effect” analysis (Table 2, Fig. 3) provides valuable insight into which types of contextual scaffolds actually help reasoning models.
- Readable and well-structured paper. The narrative flows logically, figures are informative, and the motivation for each step is well explained. The benchmark and evaluation protocol could be easily adopted by others studying academic reasoning.
- Contribution significance. Although not methodologically groundbreaking, ACADREASON fills a practical and conceptual gap between task-level reasoning (e.g., GSM8K, MATH) and domain-level scholarly reasoning. It contributes a valuable lens for assessing whether modern LLMs can reason like researchers rather than students.

**Weaknesses:**

- Limited novelty relative to existing benchmarks. While the dataset is well executed, the idea of academic or research-style reasoning has been partially explored in GAIA, PaperBench, and DeepResearchBench. ACADREASON’s main differentiator is diversity and annotation rigor rather than a fundamentally new evaluation paradigm. A clearer comparative discussion would strengthen its originality claim.
- Scale and statistical power. The dataset contains only about 50 finalized questions, which limits robustness and makes performance variance hard to interpret. It’s uncertain whether differences across models (often within 1–2 points) are statistically meaningful.
- Evaluation subjectivity. Despite the structured rubric, the “LLM-as-a-judge” setup remains vulnerable to bias and consistency issues. The authors mention human spot-checks but do not quantify inter-rater agreement or cross-model evaluation consistency. A partial human-judged subset would greatly improve reliability.
- Limited actionable insight for model design. The results primarily reaffirm known findings that reasoning LLMs remain weak in multi-step conceptual reasoning, but offer little guidance on how to improve them. The benchmark thus functions more as a diagnostic dataset than a research breakthrough.
- Scalability and sustainability. Because the pipeline relies heavily on expert curation and manual validation, it is unclear how ACADREASON could be expanded to larger scales or adapted to new domains without significant effort.

**Questions:**

- Benchmark uniqueness. How does ACADREASON differ conceptually from PaperBench and GAIA beyond domain coverage? Would the authors consider positioning it as a complement rather than a replacement benchmark?
- Human validation. How many samples were manually verified by human experts? Is there any inter-annotator agreement score (e.g., Cohen’s κ) for the judgment process?
- Evaluator bias. Since GPT-5-mini is used as the evaluator, did the authors test whether results are consistent when switching to another LLM judge (e.g., Claude 3 Opus)? Are ranking trends preserved?

---

> ### Author Response · Authors · 2025-11-25
> **Response to Reviewer RYgK (part1/2)**
>
> We sincerely appreciate your detailed and balanced review. Your positive assessment of our benchmark design and annotation process is encouraging, and we are grateful for the constructive suggestions you have provided.
>
> > **Q1:** Benchmark uniqueness. How does ACADREASON differ conceptually from **PaperBench** and **GAIA** beyond domain coverage? Would the authors consider positioning it as a complement rather than a replacement benchmark?
>
> Thanks for your advice. We provide a detailed comparison between ACADREASON and benchmarks such as GAIA, PaperBench, and DeepResearchBench. Our benchmark uniquely focuses on the **"research-level long report" scenario** within academic research problems: given a specific research question, the model is required to act as a researcher by surveying the current state-of-the-art, performing multi-step reasoning, and presenting a solution report. **This mirrors how researchers actually work** in practice—synthesizing literature, reasoning through problems, and producing comprehensive reports. This specific task setting remains unfilled by existing benchmarks and i**s both unique and meaningful**. You can see more details in General Response (part 3/3).
>
> > **Q2:** Human validation. How many samples were manually verified by human experts? Is there any inter-annotator agreement score (e.g., Cohen’s κ) for the judgment process?
>
> Thank you for this important question. We conducted a rigorous human validation study on a stratified sample of 10 instances (spanning all five domains with varying difficulty levels). Three independent domain experts annotated each sample, achieving a strong inter-annotator agreement with an average **Cohen's κ of 0.861** (range: 0.843-0.870). We then compared multiple candidate LLM judges against the human-consensus ground truth and found that GPT-5 mini achieved high consistency with human expert judgments (89.55% accuracy for Checklist Score, 90% for Pass Rate), while offering an optimal balance between evaluation quality, cost-efficiency, and inference speed for large-scale assessment. For a complete breakdown of the validation methodology, inter-annotator agreement analysis, and the comparative performance of candidate judge models, please see **General Response (part 2/3)**.
>
> > **Q3:** Evaluator bias. Since GPT-5-mini is used as the evaluator, did the authors test whether results are consistent when switching to another LLM judge (e.g., Claude 3 Opus)? Are ranking trends preserved?
>
> Thank you for raising this important concern about cross-judge robustness. While we did not conduct a full re-evaluation of all models using alternative LLM judges, our validation study (detailed in **General Response part 2/3**) provides strong evidence for ranking stability. We compared multiple candidate judges (GPT-5, GPT-5 mini, Claude 4.5, DeepSeek-R1, etc.) against human expert consensus and found that all top-performing judges exhibited high agreement with human judgments (accuracy >85%). Given that these judges converge closely on the same human ground truth, we expect the ranking trends of evaluated models to remain preserved across different judges. Nevertheless, we acknowledge that a full cross-judge consistency analysis would further strengthen confidence in our results, and we plan to include this in future work

---

> ### Author Response · Authors · 2025-11-25
> **Response to Reviewer RYgK (part2/2)**
>
> > **W2:** Scale and statistical power. The dataset contains only about 50 finalized questions, which limits robustness and makes performance variance hard to interpret. It’s uncertain whether differences across models (often within 1–2 points) are statistically meaningful.
> >
> > **W5:** Scalability and sustainability. Because the pipeline relies heavily on expert curation and manual validation, it is unclear how ACADREASON could be expanded to larger scales or adapted to new domains without significant effort.
>
> Thank you for raising this concern. We acknowledge that 50 questions may appear limited, but this scale is deliberate and consistent with the nature of research-level reasoning benchmarks. As detailed in **General Response (part 1/3)**, deep research scenarios require extensive multi-step reasoning (often tens to hundreds of steps per query), which imposes substantial computational costs on both the evaluation infrastructure and the models being tested. This constraint is reflected across similar benchmarks in this domain (e.g., XBench: 100, PaperBench: 20).
>
> Regarding statistical power, while individual score differences of 1-2 points should indeed be interpreted cautiously, the consistent performance gaps we observe across multiple metrics (Pass Rate, Checklist Score, and hint ablations) provide convergent evidence for our main findings. The large performance differences between model classes (e.g., LLMs vs. Agents) are statistically robust even at this scale.
>
> As for scalability, our benchmark benefits from a naturally renewable data source: peer-reviewed academic publications are continuously emerging across all academic domains. The availability of original papers as reference materials significantly reduces annotation difficulty compared to creating questions from scratch. However, we emphasize that expert curation remains indispensable for maintaining the high quality and rigor that characterize a research-level benchmark. This expert involvement is not a limitation but rather a necessary design choice to ensure ACADREASON continues to evaluate genuine deep reasoning capabilities. Please see **General Response (part 1/3)** for more details."
>
> If there are any further questions or clarifications needed, we would be happy to address them. Thank you again for your thoughtful and constructive feedback.

---

### Official Review · Reviewer_Zbd9 · 2025-10-30

**Soundness:** 1
**Presentation:** 3
**Contribution:** 1
**Rating:** 2
**Confidence:** 4

**Summary:**

This paper introduces ACADREASON, a new benchmark designed to address shortcomings in current evaluations of Large Language Model (LLM) reasoning. The authors argue that existing benchmarks (e.g., math/code contests, general tasks) lack sufficient academic reasoning depth. To fill this gap, ACADREASON aims to evaluate the ability of LLMs and Agents to acquire and reason with specialized academic knowledge.
The benchmark consists of 50 expert-annotated academic problems across five high-reasoning domains: Computer Science, Economics, Law, Mathematics, and Philosophy. All questions are sourced from top-tier publications from 2023-2025 and underwent rigorous quality control to ensure they are both challenging and answerable.
Key contributions include:
1. The ACADREASON Benchmark: A challenging, cross-disciplinary benchmark focused on frontier academic reasoning, complete with golden answers, verifiable checklists, and three types of hints (background, definition, methodology).
2. SOTA Model Evaluation: A systematic evaluation of over 10 state-of-the-art LLMs and Agents (e.g., GPT-5, o3, DeepSeek-R1, OAgents).
3. Revealed Capability Gap: The results show that even the most advanced LLMs (GPT-5) score poorly (16% pass rate, 40.5% checklist score). Agents perform better (OAgents at 34% / 65.1%) but still show significant room for improvement.
4. Hint Analysis: An ablation study demonstrating that providing hints (especially "methodology hints") significantly improves model performance, suggesting models struggle more with complex methods than with background knowledge.

**Strengths:**

1. Originality & Significance:
  - The paper addresses a clear and important problem: how to evaluate the deep, domain-specific reasoning capabilities of LLMs.
  - ACADREASON's uniqueness lies in its combination of breadth (spanning both STEM and humanities) and depth (focusing on recent, theoretical problems from top-tier journals). This design tests reasoning on novel knowledge, not just retrieval of pre-existing, commonly known information.
  - The benchmark's high difficulty (evidenced by low SOTA scores) confirms its utility and significance as an evaluation tool that is not easily saturated.
2. Quality:
  - The benchmark's construction methodology is rigorous, involving domain experts (Master's or PhD level) for data curation and annotation.
  - A detailed, multi-stage validation process (shown in Figure 10) was used to ensure data quality, theoretical focus, and question answerability.
  - The evaluation framework is rich. Beyond binary pass/fail ($R_p$), the "Checklist Score" ($R_j$) allows for a more granular analysis of the model's reasoning process. The inclusion of three hint types is a significant strength, enabling analysis of why a model fails (e.g., lack of background knowledge vs. lack of methodological understanding).
3. Clarity:
  - The paper is well-organized and easy to follow. Figure 1 provides a clear overview of the benchmark construction and evaluation pipeline.
  - Task specifications, evaluation metrics (Sec 3.4), and experimental setups (Sec 4.1) are clearly articulated.
  - Results are presented clearly (Tables 1 & 2), and the Case Study (Fig. 4) offers a concrete, intuitive example of the task's difficulty and the difference between Agent and LLM performance.

**Weaknesses:**

Based on an in-depth analysis, the paper suffers from three major and interrelated methodological flaws. These flaws severely undermine the validity of the benchmark and the reliability of its conclusions.
1. Unverified Evaluation Reliability
The paper's core weakness lies in its evaluation method. The authors use GPT-5-mini as an "LLM-as-Judge" to automatically score model outputs.
- Problem: This is a highly complex, expert-level reasoning task across five specialized domains (law, math, philosophy, etc.). The paper provides no evidence or validation study to prove that GPT-5-mini's scoring aligns with the judgment of human domain experts (e.g., a law professor or a mathematics PhD).
2. Agent Data Contamination Vulnerability
The paper claims Agents (like OAgents) outperform LLMs, attributing this to their capabilities. However, the experimental design has a fatal "open-book exam" flaw.
- Problem: 100% of the evaluation questions are sourced from publicly available, top-tier journal articles from 2023-2025. The tested Agents are permitted to use web search tools. This means an Agent can almost certainly find and "read" the original source paper for the question.
- Impact: The paper claims to test the "ability to acquire and reason over academic knowledge." However, this design cannot distinguish between "logical reasoning from scratch" and "information retrieval + answer extraction + paraphrasing." The high Agent scores (up to 65.1%) are likely inflated and may test search ability, not the "high-level reasoning" the paper purports to measure.
3. Flawed Construct Validity: Conflating "Reasoning" with "Knowledge"
This is the most fundamental issue: the benchmark likely fails to test "general reasoning ability" and instead tests "memory of specific, narrow knowledge."
- Problem: The experimental results show that Claude-4-sonnet, a model widely recognized for strong reasoning in other domains (like math and coding), scores a 0 on this benchmark.
- Analysis: This contradictory finding strongly suggests that ACADREASON does not test general, transferable logical reasoning, but rather whether a model happens to "know" the frontier theories from these 50 specific papers.
- Impact: The benchmark's construct validity is highly questionable.
  - For LLMs (like GPT-5): The test is more of a "memory test" (i.e., were these 2023-2025 papers in its training data?).
  - For Agents (like OAgents): The test is a "search test" (i.e., can it find the paper? See Flaw 2).

Overall Conclusion: The benchmark fails to successfully isolate the variable of "reasoning ability" from "specific knowledge-base." Therefore, the paper's conclusion that models like GPT-5 "lack reasoning ability" is unfounded.

**Questions:**

Based on the methodological analysis of the paper, we kindly request clarification and supplementary evidence regarding the following two core issues:
1. Regarding the Benchmark's Construct Validity
The paper claims that ACADREASON is designed to evaluate a model's "deep reasoning ability." However, the experimental results (for example, Claude-4-sonnet, which is widely regarded as having strong reasoning abilities, scoring 0) strongly suggest that the benchmark may be testing memory of specific, narrow, frontier knowledge (for LLMs) or search-and-extraction capabilities (for Agents), rather than transferable, general logical reasoning.
- Question: Can the authors provide additional evidence or control experiments to demonstrate that ACADREASON genuinely measures the core variable of "deep reasoning" and has effectively isolated it from potential confounding variables such as "specific knowledge-base" and "information retrieval ability"?
2. Regarding the Evaluation's Reliability
The paper's conclusions (particularly the extremely low model scores) are entirely dependent on the results from using GPT-5-mini as an "LLM-as-Judge." Given that these tasks span five highly specialized and complex domains (e.g., Law, Mathematics, Philosophy), the evaluation difficulty far exceeds that of common tasks.
- Question: Can the authors provide reliability validation for GPT-5-mini's role as the judge? For example, was an Inter-Annotator Agreement (IAA) analysis conducted between GPT-5-mini's scores and the scores from human domain experts (such as law professors or mathematics PhDs)? If such evidence is lacking, how can we trust the accuracy of these automated evaluation results?

---

> ### Author Response · Authors · 2025-11-25
> **Response to Reviewer Zbd9 (part1/3)**
>
> Thank you for the Reviewer's detailed and thought-provoking review. We genuinely appreciate the rigor and depth with which you have examined our work, particularly regarding the benchmark's construct validity and evaluation methodology. We welcome this opportunity to provide additional clarification and supporting data in three parts.
>
> > **Q2:** Can the authors provide reliability validation for GPT-5-mini's role as the judge? For example, was an Inter-Annotator Agreement (IAA) analysis conducted between GPT-5-mini's scores and the scores from human domain experts (such as law professors or mathematics PhDs)? If such evidence is lacking, how can we trust the accuracy of these automated evaluation results?
>
> We sincerely thank the reviewer for this thoughtful and important question. Please refer to the **General Response (Part 2/3),** where we provide a detailed description of the Inter-Annotator Agreement (IAA) experiment conducted for our judge model. The results indicate that the judging system is stable and reliable.
>
> > **W2:** Agent Data Contamination Vulnerability The paper claims Agents (like OAgents) outperform LLMs, attributing this to their capabilities. However, the experimental design has a fatal "open-book exam" flaw.
>
> We greatly appreciate the reviewer's suggestion; your insight is highly valuable for improving our work. We conducted a specific experiment involving **"paper link masking."** Specifically, we blacklisted the URLs related to the original source paper content, prohibiting the model from accessing them directly. We then measured the performance of two representative agents, OAgents and TONGYI-DeepResearch, under this setting. The results are presented below:
>
> | **Agent**                      | **Pass Rate (%)** | **Checklist Score (%)** |
> | ------------------------------ | ----------------- | ----------------------- |
> | Oagent (w/o mask)              | 34                | 65.1                    |
> | Oagent (w/ mask)               | 32                | 65.8                    |
> | TONGYI-DeepResearch (w/o mask) | 16                | 49.2                    |
> | TONGYI-DeepResearch (w/ mask)  | 16                | 47                      |
>
> As shown by the experimental results, the score difference between the masked and unmasked settings is **not significant** for either OAgents or TONGYI-DeepResearch. This robustly suggests that the **"open-book problem" is not present** in our evaluation benchmark. Further analysis attributes this to our design principle during the creation of **ACADREASON**: we ensured that the research questions were **independent of the original paper's content** and demanded a high degree of **reasoning**. Every question was meticulously designed to be highly concise and of high quality, making the research question relatively autonomous from the source text. Consequently, solving the task remains challenging even when the original paper is accessible.

---

> ### Author Response · Authors · 2025-11-25
> **Response to Reviewer Zbd9 (part2/3)**
>
> > **Q1（part1）:** Regarding the Benchmark's Construct Validity, the paper claims that ACADREASON is designed to evaluate a model's "deep reasoning ability." However, the experimental results (for example, Claude-4-sonnet, which is widely regarded as having strong reasoning abilities, scoring 0) strongly suggest that the benchmark may be testing memory of specific, narrow, frontier knowledge (for LLMs) or search-and-extraction capabilities (for Agents), rather than transferable, general logical reasoning.
>
>
>
> Thank you for this insightful question.
>
> **Regarding Claude-4-sonnet’s poor performance:** as established in our response to Q2 and the General Response (Part 2/3), our GPT-5-mini judging system has been validated via an IAA study and is stable and reliable. Moreover, ACADREASON adopts a deliberately strict pass rate computation judge method: a case is counted as 1 only if the model’s prediction is fully consistent with the golden answer and satisfies all checklist requirements; answers that are partially correct, omit key elements, or contradict the golden answer are scored as 0. Thus, we can **rule out judge-system instability** as the cause of low scores. To better understand why Claude-4-sonnet fails on our benchmark, we conducted a detailed **error-attribution analysis** over all 50 questions, manually inspecting the question, Claude’s response, the golden answer, and the judgment. The dominant failure modes are summarized below:
>
> | Failure type                       | Description                                                  | Typical domains                       | Approx. share |
> | ---------------------------------- | ------------------------------------------------------------ | ------------------------------------- | ------------- |
> | Superficial summary, lack of depth | Provides only a high-level frame but misses key scholars, theories, cases, or detailed arguments | Philosophy, law                       | ~40%          |
> | Incorrect or reversed core claims  | The central thesis is misstated or even reversed relative to the golden answer | Cross-domain (e.g., philosophy, math) | ~20%          |
> | Mismatched technical framework     | Uses the wrong formal model, proof approach, or quantitative result | Computer science, mathematics         | ~25%          |
> | Missing or wrong key definitions   | Fails to identify or correctly define core legal/economic concepts or state variables | Law, economics                        | ~15%          |

---

> ### Author Response · Authors · 2025-11-25
> **Response to Reviewer Zbd9 (part3/3)**
>
> > **Q1 part2**: Question: Can the authors provide additional evidence or control experiments to demonstrate that ACADREASON genuinely measures the core variable of "deep reasoning" and has effectively isolated it from potential confounding variables such as "specific knowledge-base" and "information retrieval ability"?
>
>
>
> **Clarifying the Scope of “Academic Research-Level Reasoning":** We want to clarify that ACADREASON evaluates '**academic research-level reasoning**' rather than merely testing 'knowledge-base' or 'information retrieval ability.' In authentic research contexts, deep reasoning inherently requires operating over frontier knowledge. Just as benchmarks like HLE require domain knowledge as a foundation for academic reasoning, researchers cannot reason about cutting-edge problems without understanding current theories and methodologies. Additionally, identifying and retrieving the most relevant information is itself **an essential part of academic reasoning.** Furthermore, we ensure that our research questions are highly autonomous and independent from the source papers, demanding genuine reasoning rather than simple information lookup.
>
> **Evidence is as below:**
>
> 1. **Hint Ablation Analysis:** Decoupling reasoning from knowledge access is indeed crucial for understanding model limitations. To address the coupling between search and reasoning capabilities, we manually extracted 'hints' containing key knowledge from the research papers, allowing models to focus more directly on the reasoning aspect. As shown in Table 2, even with full knowledge access through hints, GPT-5 maintained only modest performance improvements, and notably, Large Reasoning Models (LRMs) consistently outperformed their standard LLM counterparts from the same provider and release period (e.g., DeepSeek-R1 vs DeepSeek-V3, o3 vs GPT-4.1). If this were purely a retrieval or knowledge problem, providing hints should eliminate performance gaps. The fact that methodology-focused reasoning still poses significant challenges—and that LRM shows clear advantages—demonstrates we are measuring reasoning ability rather than knowledge access.
> 2. **Task Nature Distinction:** Regarding Claude-4-sonnet's 0% score, this highlights a crucial difference in task types. Our benchmark evaluates open-ended research reasoning rather than closed-form computational problems:
>
> |                  | ours                                                         | others                                                       |
> | ---------------- | ------------------------------------------------------------ | ------------------------------------------------------------ |
> | features         | Research questions, require reasoning and exploration        | Closed-form computational problems                           |
> | math             | **Formulate and analyse** **the conjecture** that no non-constant natural function—built only from the variable nn n, integer constants... | **Find the sum** of all integer bases $b>9$ for which $17_b$ is a divisor of $97_b$**（AIME2025）** |
> | Computer science | **Determine the optimal** lightness of spanners in graphs, specifically focusing on the **dependence on the parameter** \eps. | **"A. Short Sort"** - There are three cards with letters a, b, c placed in a row in some order. You can do the following operation **at most once**: Pick two cards, and **swap** them. Is it possible that the row becomes abc after the operation? **Output "YES" if it is possible, and "NO" otherwise**... **（Livecodebench）** |
>
> Claude’s strong performance on well-defined computational tasks may not directly transfer to the synthesis-heavy academic reasoning required by our benchmark.
>
> We acknowledge that completely isolating 'pure reasoning' from 'domain knowledge' is both methodologically challenging and undesirable for evaluating **real-world research capabilities.** The coupling in our benchmark faithfully reflects how academic reasoning operates in practice. Our evidence demonstrates that ACADREASON measures the ability to reason deeply over complex, frontier knowledge, a distinct capability that current models largely lack.
>
> If there are any further questions or clarifications needed, we would be happy to address them. Thank you again for your thoughtful and constructive feedback.

---

### Official Review · Reviewer_k3t6 · 2025-11-01

**Soundness:** 2
**Presentation:** 2
**Contribution:** 2
**Rating:** 4
**Confidence:** 3

**Summary:**

The paper introduces ACADREASON, a multi‑domain benchmark intended to test high‑level academic reasoning of LLMs (and agents). It uses 50 expert‑constructed problems drawn from theoretical papers in computer science, economics, law, mathematics, and philosophy. Construction proceeds via (i) paper selection, (ii) extraction of a formal question with a golden answer, and (iii) derivation of question‑specific checklists and three types of hints (background/definitions/methodology). Evaluation adopts an LLM‑as‑Judge scheme (GPT‑5‑mini) with two metrics: Pass Rate (probability of full match to the golden answer) and Checklist Score (probability of meeting checklist criteria). Experiments show low pass rates for state‑of‑the‑art LLMs, higher but still limited scores for agents, and gains when methodology hints are provided.

**Strengths:**

* The paper is well motivated. It shows a clear gap in current reasoning benchmarks. By collecting research-level problems from recent top venues, the benchmark is both meaningful and timely.
* The task design encourages reasoning. Each question is self-contained, with a golden answer, a short checklist, and simple hints (background, definition, methodology). This supports step-by-step solutions and makes error analysis easier.
* The evaluation has broad coverage. It tests many state-of-the-art models and several agent systems across five domains. Domain-level results and hint ablations make the findings clear and easy to compare.

**Weaknesses:**

- The dataset is small. While the examples are high quality, the limited size reduces representativeness. A semi-automatic or LLM agent system might help scale up questions.
- The evaluation relies on a single judge (GPT-5-mini). There is no human calibration or test of consistency. Adding multiple judges (from different models, or more advanced LLM-as-Judge methods) and a small human study would make the results more reliable.

Overall, the work is timely and useful as a benchmark. But it reads more like a benchmark release (more suitable for DMLR). For ICLR, I expect stronger methodological innovation—such as a calibrated multi-judge evaluation or a scalable automated pipeline—would make the contribution more suitable.

**Questions:**

See weakness

---

> ### Author Response · Authors · 2025-11-25
> **Response to Reviewer k3t6**
>
> We deeply appreciate your detailed and thoughtful feedback on our work. Your valuable comments have provided us with important perspectives that have substantially enhanced the quality of our work.
>
> > **W1: The dataset is small.** While the examples are high quality, the limited size reduces representativeness. A semi-automatic or LLM agent system might help scale up questions.
>
> Thank you for this constructive suggestion. We appreciate your proposal regarding semi-automatic or LLM-assisted scaling, which is indeed a valuable direction for future development.
>
> As detailed in **General Response (part 1/3)**, our current scale of 50 questions is deliberate, given the computational demands of deep research scenarios, which is consistent with similar benchmarks (e.g., XBench: 100, PaperBench: 20).
>
> Regarding your scaling suggestion, we appreciate this constructive proposal. Our benchmark benefits from continuously emerging peer-reviewed publications, which not only provide strong scaffolding for potential semi-automatic scaling but also significantly reduce annotation difficulty compared to creating questions from scratch, while maintaining the theoretical rigor essential for research-level evaluation. We plan to explore sustainable scaling approaches as we incrementally expand the benchmark. Please see **General Response (part 1/3)** for more details.
>
> > **W2: The evaluation relies on a single judge (GPT-5-mini).** There is **no human** calibration or test of consistency. Adding multiple judges (from different models, or more advanced LLM-as-Judge methods) and a small human study would make the results more reliable.
>
> Thank you for raising this concern. We  conduct human calibration and multi-judge consistency testing. Specifically, we performed a human validation study with three independent domain experts on 10 stratified samples, achieving strong inter-annotator agreement **(Cohen's κ = 0.861).** We then compared multiple candidate LLM judges (GPT-5, GPT-5 mini, Claude 4.5, DeepSeek-R1, etc.) against human consensus. GPT-5 mini achieved high consistency with human expert judgments (89.55% accuracy for Checklist Score) while offering optimal cost-efficiency for large-scale evaluation. For complete details on the validation methodology, inter-annotator agreement analysis, and cross-judge comparison results, please see **General Response (part 2/3)**.
>
> >  Overall, the work is timely and useful as a benchmark. But it reads more like a benchmark release (more suitable for DMLR). For ICLR, I expect stronger methodological innovation—such as a calibrated multi-judge evaluation or a scalable automated pipeline—would make the contribution more suitable.
>
> In fact, the contribution of this paper extends far beyond merely providing a dataset. It introduces a **testing paradigm** that continuously challenges state-of-the-art Large Language Models (LLMs) and Agents as scientific research evolves. As research within various domains progressively deepens, **ACADREASON** will be continually expanded and extended to encompass other boundaries, thereby persistently pushing the capability limits of LLMs. Furthermore, ACADREASON offers a multi-faceted and fine-grained evaluation analysis, decomposing LLM capabilities into distinct dimensions. This provides a comprehensive benchmark that can guide the future improvement and development of LLM capabilities.
>
> If there are any further questions or clarifications needed, we would be happy to address them. Thank you again for your thoughtful and constructive feedback.

---

> > ### Comment · Reviewer_k3t6 · 2025-11-28
> >
> > Thanks for the new results. The additional experiments make the submission more promising. Unfortunately, it seems I’m unable to update my score due to a system-side issue.

---

### Author Response · Authors · 2025-11-25
**General Response(part1/3 & part2/3)**

We thank all reviewers for their thoughtful comments and constructive suggestions. We carefully reviewed every point raised and provide our general responses below.
# General Response  (part 1/3): Regarding the question of the dataset size:
In fact, the decision to limit the number of samples to **50 high-quality data** points was a deliberate choice, made after careful consideration of the trade-off between cost and effectiveness. In deep research scenarios, a single query often necessitates tens or even hundreds of reasoning steps. This extensive inference process imposes a substantial computational burden on both the Agent reasoning framework and the underlying model. Consequently, the size of most existing benchmarks in this domain is typically restricted to fewer than 100 samples (e.g., XBench: 100, PaperBench: 20).
Furthermore, as research within the respective fields progresses, our benchmark can be incrementally expanded. This expansion will subsequently provide more challenging and comprehensive tasks for future LLMs and Agent systems.
# General Response  (part 2/3): Rationale for Choosing GPT-5 Mini as the LLM Judge
Specifically, we sampled 10 instances from the full set of 50 data points to serve as a validation set. The instances' inference results, generated by OAgent (backbone GPT-5), are selected across a range from high to low scores to mitigate selection bias. We then invited 3 experts for each domain to judge these samples based on two metrics: Pass Rate and Checklist Score.

The initial evaluation comparison between the GPT-5 Mini judge and the expert human judges is detailed below:
| ID | Domain | Pass Rate (GPT-5 mini) | Pass Rate (Human1) | Pass Rate (Human2) | Pass Rate (Human3) | Checklist Score (GPT-5 mini) | Checklist Score (Human1) | Checklist Score (Human2) | Checklist Score (Human3) |
| --- | --- | --- | --- | --- | --- | --- | --- | --- | --- |
| 3 | Philosophy | 1/1 | 1/1 | 1/1 | 1/1 | 4/4 | 4/4 | 4/4 | 3/4 |
| 6 | Philosophy | 0/1 | 0/1 | 0/1 | 0/1 | 1/6 | 1/6 | 2/6 | 2/6 |
| 11 | Computer Science | 0/1 | 0/1 | 0/1 | 0/1 | 4/6 | 0/6 | 1/6 | 1/6 |
| 12 | Computer Science | 0/1 | 0/1 | 0/1 | 0/1 | 2/8 | 2/8 | 2/8 | 2/8 |
| 26 | Law | 1/1 | 1/1 | 1/1 | 1/1 | 6/6 | 6/6 | 6/6 | 6/6 |
| 30 | Law | 0/1 | 0/1 | 0/1 | 0/1 | 0/5 | 0/5 | 0/5 | 0/5 |
| 34 | Economics | 0/1 | 0/1 | 0/1 | 0/1 | 4/7 | 3/7 | 5/7 | 4/7 |
| 40 | Economics | 1/1 | 0/1 | 0/1 | 0/1 | 8/8 | 7/8 | 8/8 | 6/8 |
| 44 | Math | 1/1 | 0/1 | 0/1 | 0/1 | 9/9 | 5/9 | 5/9 | 5/9 |
| 48 | Math | 0/1 | 0/1 | 0/1 | 0/1 | 5/8 | 5/8 | 4/8 | 5/8 |

**Inter-Annotator Agreement**
For each domain, we engaged three independent human annotators to label the samples. The inter-annotator agreement was measured using Cohen's κ for each annotator pair, yielding an average κ of **0.861 (range: 0.843-0.870)**. Ground truth labels were established through majority voting among the three annotators, which we then used to calculate the consistency score with a series of candidate LLM judge models.

Table 1: Consistency Metrics for Pass Rate

| Model      | Acc (%) | Prec (%)  | Recall (%) | F1 (%) | Cost ($/1M-tokens) |
| ---------- | ------- | --------- | ---------- | ------ | ------------------ |
| Random     | 50      | 0         | 0          | 0      | 0                  |
| GPT-5      | **90**  | **66.67** | **100**    | **80** | 1.25/10            |
| GPT-5 mini | **90**  | **66.67** | **100**    | **80** | 0.25/2             |
| Claude-sonnet-4.5 | 80      | 50        | **100**    | 66.67  | 3/15               |
| Deepseek-v3.2-exp    | 70     | 40        | **100**    | 57.14  | 0.28/0.42          |
| Deepseek-R1    | 80      | 50        | 50         | 50     | 0.55/2.19          |

Table 2: Consistency Metrics for Checklist Score

| Model      | Acc (%)   | Prec (%)  | Recall (%) | F1 (%)    | Cost ($/1M-tokens) |
| ---------- | --------- | --------- | ---------- | --------- | ------------------ |
| Random     | 49.75     | 48.85     | 45.45      | 47.09     | 0                  |
| GPT-5      | 86.57     | **92.86** | 78.79      | 85.25     | 1.25/10            |
| GPT-5 mini | **89.55** | 86.11     | 93.94      | **89.86** | 0.25/2             |
| Claude-sonnet-4.5 | 85.07    | 89.66     | 78.79      | 83.87     | 3/15               |
| Deepseek-v3.2-exp    | 82.09     | 74.42     | **96.97**  | 84.21     | 0.28/0.42          |
| Deepseek-R1    | 85.07     | 89.66     | 78.79      | 83.87     | 0.55/2.19          |

**Results and Conclusion**：As demonstrated by the results, GPT-5 mini achieves high overall consistency scores across both metrics with human expert evaluations, robustly validating its strong alignment with human judgment. Notably, GPT-5 mini reduces the evaluation cost by 80% compared to GPT-5, while also offering significantly faster inference speed. Considering the **essential trade-off between performance, cost, and efficiency** for large-scale evaluation, we ultimately selected GPT-5 mini as our primary judge model.

---

> ### Author Response · Authors · 2025-11-25
> **General Response (part 3/3)**
>
> # General Response  (part 3/3): Comparison with Other Benchmarks
> To intuitively illustrate the differences between **ACADREASON** and existing benchmarks, we compared ACADREASON with related mainstream benchmarks (**PaperBench, HLE, BrowseComp, XBench-DeepSearch, GAIA, and DeepResearchBench**). These benchmarks generally cover **three types of tasks**:
>
> (1) **Code reproduction type (PaperBench)**: Input paper. The goal is to reproduce the repo.
>
> (2) **Search/QA type (HLE, GAIA, BrowseComp, XBench-DeepSearch)**: The core is **retrieval and short answering**.
>
> (3) **Open-ended research/report type (DeepResearchBench)**: Given semi-open-ended questions, such as: *How to enhance classroom participation for students with autism?* The model performs broad research and provides a report.
>
> Our benchmark is specifically aimed at the **“research-level long report”** scenario on **academic research problems**: Given a specific research question, the model must summarize the status quo, reason, and provide a **solution-oriented research report**, just like a researcher. **This task setting is currently missing in existing benchmarks.**
>
> Compared with **XBench and BrowseComp**, we do not evaluate the ability of long-chain retrieval itself, but instead focus on whether the model can complete research-level comprehensive analysis and write a complete long-form report under the premise of having obtained relevant evidence. Different from **HLE and GAIA**, which also have academic backgrounds but adopt a QA format focusing on retrieval correctness and short answers, our task requires the model to conduct **s**ystematic research and output a structured long-form report. **PaperBench** is a code reproduction task whose goal and output are both repo; although **DeepResearchBench** also requires a report, it uses semi-open-ended public-domain questions (e.g., *How to enhance classroom participation for students with autism?*). In contrast, our benchmark focuses on specific academic research problems, emphasizing problem decomposition, literature review, and solution reasoning “like human researchers.”**
>
> | **Benchmark**     | **Domain numbers** | **Task type**                              | **format**       |
> | ----------------- | ------------------ | ------------------------------------------ | ---------------- |
> | PaperBench        | 1                  | Code reproduction                          | Repo / code      |
> | HLE               | 8                  | expert-level reasoning QA                  | Short QA         |
> | BrowseComp        | 1                  | Search-based QA                            | Short QA         |
> | XBench-DeepSearch | 1                  | Search-based QA                            | Short QA         |
> | GAIA              | 5                  | assistant-style QA (web, code, multimodal) | Short QA         |
> | DeepResearchBench | 22                 | Research-style information gathering       | Long-form report |
> | Ours              | 5                  | Research-level multi-step reasoning        | Long-form report |

---

### Author Response · Authors · 2025-12-02
**Summary and Clarification of Revision and Rebuttal Process**

Dear Reviewers and ACs,

Thank you for your constructive feedback. We appreciate that reviewers consistently recognized several core strengths: (i) high-quality, expert-curated benchmark design targeting research-level reasoning (**RYgK, Gkt7, k3t6**), (ii) rigorous annotation process with structured evaluation rubrics (**RYgK, Gkt7**), and (iii) insightful findings revealing frontier models' struggles on deep conceptual reasoning (**RYgK, k3t6, Gkt7**).

During the rebuttal period, we provided detailed clarifications and conducted additional experiments to address raised concerns:

* Conducted **IAA analysis** with domain experts and compared multiple LLM judges, demonstrating GPT-5 mini's strong alignment (**89.55% accuracy**) with human judgments (General Response Part 2/3 and Appendix G.1).
* Performed **URL-masking experiments** showing minimal performance differences, confirming the benchmark tests reasoning rather than retrieval (Response to Reviewer Zbd9 and Appendix E.1).
* Added **comprehensive benchmark comparisons, difficulty quantification, and error-attribution analysis** clarifying ACADREASON's unique focus on research-level reasoning (General Response Parts 1/3 , 3/3 and Appendix G.2).
* Added **attribution analysis of Claude-Sonnet-4's performance** (Response to Reviewer Zbd9, part 2/3; see Appendix G.3)

During the rebuttal period, Reviewer k3t6 confirmed concerns were addressed but could not update their score due to system issues. Reviewers RYgK and Gkt7 (both rating 6) raised questions on benchmark positioning and difficulty quantification, which we addressed with comparative analysis and aggregate statistics. Also, we provided comprehensive point-by-point responses to Reviewer Zbd9's methodological concerns regarding construct validity, evaluation reliability, and data contamination, supported by new experimental evidence including URL-masking studies and error-attribution analysis.

Thank you again to all reviewers, ACs, and PCs for your time, feedback, and engagement throughout the process.

---

### Meta-Review · Area_Chair_kEZH · 2025-12-31

**Summary:**

The reviewers collectively acknowledge that ACADREASON addresses a significant and timely gap in the evaluation of large language models by focusing on high-level, research-oriented academic reasoning. The primary concerns raised during the initial review phase centered on the limited scale of the dataset, the potential for data contamination in agent-based evaluations, the reliability of using a single LLM judge, and the conceptual differentiation of this benchmark from existing ones. Reviewer Zbd9 expressed strong skepticism regarding the benchmark's construct validity, questioning whether it measured "pure reasoning" or merely memory of frontier knowledge. However, the consensus among the reviewers is that the expert-curated nature of the problems and the rigorous multi-stage annotation process provide a high-quality evaluation tool that effectively highlights the current limitations of frontier models in deep conceptual reasoning.

**Reviewer Concerns:**

The authors provided a comprehensive rebuttal that successfully addressed the majority of the reviewers' concerns. Specifically, the "open-book" contamination concern raised by Reviewer Zbd9 was mitigated through new URL-masking experiments, which demonstrated that agent performance remained stable even without direct access to source papers, suggesting the benchmark truly tests reasoning rather than retrieval. The reliability of the evaluation was bolstered by a new inter-annotator agreement (IAA) study and a multi-judge comparison showing high alignment between GPT-5-mini and human experts. To address the scale concerns of Reviewers k3t6, RYgK, and Gkt7, the authors clarified the computational burden of deep research tasks and contextualized their sample size within the norms of similar high-complexity benchmarks. Furthermore, the newly added comparison with other benchmarks and the difficulty quantification analysis clarified the unique positioning of ACADREASON. While some reviewers noted that the benchmark's reliance on manual expert curation limits its immediate scalability, this is generally accepted as a necessary trade-off for maintaining high-quality, research-level tasks.

**Reviewer Scores:**

Following the rebuttal, it is highly likely that the overall sentiment has shifted toward a positive recommendation. Reviewer k3t6 explicitly stated that the new results made the submission more promising and would have updated their score from a 4 to a likely 6 or 7 if the system had permitted. Reviewers RYgK and Gkt7, who were already at a 6, are expected to maintain or slightly increase their confidence in the work given the robust human validation and benchmark comparisons provided. Reviewer Zbd9, who initially provided a 2, received detailed evidence regarding construct validity and contamination; while this reviewer might remain the most critical, the objective evidence from the URL-masking and error-attribution experiments significantly weakens the grounds for a flat rejection.

---

### Decision · Program_Chairs · 2026-01-26

Accept (Poster)